

# Towards Multiscale Modeling of Ocean Surface Turbulent Mixing Using Coupled MPAS-Ocean v6.3 and PALM v5.0

Qing Li[1] and Luke Van Roekel[1]

[1]Fluid Dynamics and Solid Mechanics, Los Alamos National Laboratory, Los Alamos, New Mexico, USA

**Correspondence:** Qing Li (qingli@lanl.gov)

**Abstract.** A multiscale modeling approach for studying the ocean surface turbulent mixing is explored by coupling an ocean general circulation model (GCM) MPAS-Ocean with the PArallel Large eddy simulation Model (PALM). The coupling approach is similar to the superparameterization approach that has been used mostly to represent the effects of deep convection in atmospheric GCMs. However, since the emphasis here is on the small-scale turbulent mixing processes and their interactions with the larger-scale processes, a high-fidelity, three-dimensional large eddy simulation (LES) model is used, in contrary to a simplified process-resolving model with reduced physics or reduced dimension commonly used in the superparameterization literature. To reduce the computational cost, a customized version of PALM is ported on the general-purpose graphics processing unit (GPU) with OpenACC, achieving 10-16 times overall speedup as compared to running on a single CPU. Even with the GPU-acceleration technique, superparameterization of the ocean surface turbulent mixing using high-fidelity and three-dimensional LES over the global ocean in GCMs is still computationally intensive and infeasible for long simulations. However, running PALM regionally on selected MPAS-Ocean grid cells is shown to be a promising approach moving forward. The flexible coupling between MPAS-Ocean and PALM outlined here allows further exploration of the interactions between ocean surface turbulent mixing and larger-scale processes, and development of better ocean surface turbulent mixing parameterizations in GCMs.

## 1 Introduction

Turbulent motions in the ocean surface boundary layer (OSBL) control the exchange of heat, momentum, and trace gases such as $CO_2$ between the atmosphere and ocean, and thereby affect the weather and climate. These turbulent motions are not resolved in regional and global ocean general circulation models (GCM) due to their small horizontal scales (1-100 m), short temporal scale ($10^3$-$10^5$ s) and non-hydrostatic nature. Effects of these subgrid-scale turbulent motions are commonly parameterized in GCMs using simple one-dimensional vertical mixing schemes (e.g., Large et al., 1994; Burchard et al., 2008; Reichl and Hallberg, 2018). However, significant discrepancies are found among many of such vertical mixing schemes, highlighting the uncertainties in our understanding of turbulent mixing in the OSBL (Li et al., 2019). Improvements in OSBL vertical mixing





parameterizations may require better constraints under different realistic conditions from either observations or high-resolution
and non-hydrostatic simulations where the turbulent motions are resolved, such as large eddy simulations (LES).

## 1.1 Superparameterization

Given that resolving the OSBL turbulence globally is far from being practical with the present computational resources, one
possible way to better represent the effects of those unresolved processes in GCMs is the so-called *superparameterization*
(Randall et al., 2003, 2016). The superparameterization apporach replaces simple one-dimensional parameterization schemes
by a more realistic representation of the subgrid-scale processses, usually a process-resolving sub-model embedded in each
grid cell of the coarse-resolution GCM, which attempts to resolve, rather than parameterize, the subgrid-scale processes. It
has long been used in atmosphere modeling community to represent the effects of cloud by embedding a simplified cloud-
resolving model in each grid cell of an atmospheric GCM (Grabowski and Smolarkiewicz, 1999; Khairoutdinov and Randall,
2001; Grabowski, 2004; Randall, 2013), and succeeded in simulating many phenomena that are challenging for conventional
cloud parameterizations, such as the diurnal cycle of precipitation and the Madden-Julian Oscillation (e.g., Khairoutdinov et al.,
2005; Benedict and Randall, 2009).

Since the instances of the process-resolving model at different GCM grid cells are independent of each other, massive
parallelism is possible. In addition, the embedded process-resolving model does not necessarily cover the total area of a GCM
grid cell. Therefore, the computational cost of the superparameterization approach can be significantly less than a global
process-resolving model. Even so, a compromise on the complexity and realism of the embedded process-resolving model has
to be made for computational efficiency. Given that the goal of superparameterization is to better represent the effects of small-
scale processes, which is the same as all other conventional parameterizations, the embedded process-resolving model does not
have to be as accurate as it would be in a focused process study. In fact, simplified two-dimensional cloud-resolving models
are commonly used (e.g., Grabowski and Smolarkiewicz, 1999; Khairoutdinov and Randall, 2001; Grabowski, 2004), with
some variants, e.g., using two perpendicular sets of narrow channels to partially overcome the two-dimensionality (Jung and
Arakawa, 2010). Recently a multiple-instance superparameterization approach using these simplified setup is also explored
to better represent the statistics of possible realizations and shows promising results (Jones et al., 2019). Alternatively, a
stochastic reduced model may also be used to replace the deterministic physics-based process-resolving model in the traditional
superparameterization applications (Grooms and Majda, 2013).

The literature of applying the superparameterization approach in the ocean GCMs is much less developed, perhaps because
the unresolved turbulent motions in the ocean are much smaller in scale than those in the atmosphere, therefore a global
superparameterization will require a much higher computational cost, and the "deadlock" situation described in Randall et al.
(2003) and Randall (2013) for the cloud parameterization problem may not have been encountered yet in the development of
ocean subgrid-scale parameterizations. An application of the superparameterization approach to open-ocean deep convection in
a coarse-resolution ocean GCM has been explored by Campin et al. (2011). They show that the superparameterization approach
can capture many of the important features in a non-hydrostatic high-resolution simulation of an idealized open-ocean deep
convection with much less computational cost.





For the open-ocean deep convection problem, the somewhat horizontally axisymmetric features and the relatively large vertical scale of the convective plumes justify the use of a plume-resolving model on a two-dimensional slice with the same

vertical grid as the coarse-resolution GCM in Campin et al. (2011), similar to the application of superparameterization to the cloud problem by Grabowski and Smolarkiewicz (1999) and Khairoutdinov and Randall (2001). However, three-dimensional process-resolving models may be required to faithfully represent other small-scale processes in the OSBL, especially those that are anisotropic in the horizontal (which requires an additional horizontal dimension) and have rich structures in the vertical (which requires finer vertical resolution than the GCMs), such as Langmuir turbulence (McWilliams et al., 1997). In addition,

the horizontal gradient of large-scale quantities is needed in the embedded process-resolving models to allow processes like baroclinic instability (Bachman and Taylor, 2016; Grooms and Julien, 2018), which is ignored in the applications of superparameterization in simulating both the cloud (e.g., Grabowski and Smolarkiewicz, 1999; Khairoutdinov and Randall, 2001) and open-ocean deep convection (Campin et al., 2011).

Embedding a three-dimensional process-resolving model with high-fidelity in coarse-resolution GCMs following the tradi-

tional superparameterization approach is likely to be computationally intensive and infeasible for long simulations (e.g., Parishani et al., 2017). With the recent advance in computational architecture and techniques, especially the acceleration technique using general-purpose graphics processing unit (GPU), one might wonder whether such a superparameterization approach in ocean GCMs becomes feasible. One of the goals of this paper is therefore to explore and evaluate the feasibility of such approach.

## 75   1.2   Towards a Better OSBL Parameterization

Although the idea of resolving the subgrid-scale processes in ocean GCMs with superparameterization is attractive, computational challenges may discourage its application in global climate simulations, especially if other computationally favorable solutions exist. Given the large discrepancies among different OSBL vertical mixing parameterization schemes (Li et al., 2019), there is likely some space for improvement in the conventional OSBL parameterizations. Then why even bother to explore the

superparameterization approach for ocean GCMs? As will be discussed in more detail later, this approach may provide a fruitful pathway towards a better OSBL parameterization.

Due to the scarcity of direct measurements of OSBL turbulence, conventional OSBL parameterization schemes are commonly derived from scaling analysis constrained by high-resolution simulations such as LES with various initial states and forcing conditions to cover a somewhat realistic parameter space (e.g., Li and Fox-Kemper, 2017; Reichl and Hallberg, 2018;

Reichl and Li, 2019). As shown by Li et al. (2019), many OSBL vertical mixing schemes excel at situations in which the scheme was originally derived, but less so in other situations. The problem is that in the development of a parameterization for some particular process, we often need to control other factors in order to isolate the effect of this particular process. Essentially a separation of scales is often assumed and the exploration of the parameter space is often restricted to a certain regime. Whereas in reality, processes at different scales interact with each other and the parameter space is vast. Li et al. (2019)

demonstrate a way to consistently evaluate different OSBL vertical mixing schemes over a variety of realistic surface forcing and background stratification conditions. It can potentially also be used to guide a systematic exploration of the parameter





space using LES. However, such approach ignores the interactions between the OSBL turbulence and larger-scale processes, such as submesoscale fronts and eddies, which have been shown to be important (e.g., Hamlington et al., 2014; Bachman and Taylor, 2016; Fan et al., 2018; Verma et al., 2019; Sullivan and McWilliams, 2019).

To study the interactions between the OSBL turbulence and larger-scale processes, simulations that resolve all the important processes across multiple scales are necessary. But they can be extremely computationally expensive. For example, LES that simultaneously resolves the OSBL turbulence and permits some submesoscale features requires a grid size of $O(1\ \mathrm{m})$ and a domain size of $O(10\ \mathrm{km})$ in the horizontal (e.g., Hamlington et al., 2014; Verma et al., 2019; Sullivan and McWilliams, 2019). Alternatively, the impact of large-scale processes on the small-scale OSBL turbulence can be studied by applying an externally

determined large-scale lateral buoyancy gradient in a much smaller LES domain (e.g., Bachman and Taylor, 2016; Fan et al., 2018), though at the expense of missing the feedback to the large-scales. In this sense, a superparameterized GCM may provide an intermediate approach to systematically examine the impact of large-scale processes (resolved in the GCM) on the small-scale OSBL turbulence (in LES) and vice versa, with a loose but controllable coupling between the two. Note that so far we have not explicitly defined the scale of the large-scale processes, as long as it is larger than the OSBL turbulence. Therefore, we may

use a high-resolution regional configuration or a low-resolution global configuration in the coupled framework depending on the problem to be addressed, allowing some flexibility to choose the coupling strategy. So the goal here is to better understand the processes that affect OSBL turbulent mixing in addition to the surface forcing and background stratification, which will promote further improvements of the conventional OSBL parameterizations.

## 1.3   A Multiscale Modeling Framework

In this work we are building towards a multiscale modeling framework for studying the OSBL turbulent mixing by coupling an ocean GCM, the Model for Prediction Across Scales-Ocean (MPAS-Ocean, Ringler et al., 2013; Petersen et al., 2018), with an LES, the PArallelized Large eddy simulation Model (PALM, Raasch and Schröter, 2001; Maronga et al., 2015). The primary goal is not to replace the conventional OSBL vertical mixing parameterizations by an embedded LES, as in the superparameterization approach. Instead, we seek to use the coupled MPAS-Ocean and PALM to systematically study the

behavior of OSBL turbulence under different forcing by the larger-scale processes that are resolved in the GCMs, and the potential interactions. Therefore, in contrast to the traditional superparameterization approach, the emphasis of applying the multiscale modeling framework in this study is on small-scale OSBL turbulence.

Therefore, in contrast to traditional superparameterization, in this multiscale modeling framework we are allowing the embedded LES to run on only selected GCM grid cells. Without the computational burden to run LES on every GCM grid cell, we

have the flexibility to choose the LES domain and resolution to make the process-resolving simulation as accurate as possible. A similar regional superparameterization approach using LES has recently been explored for the cloud problem (Jansson et al., 2019). Aside from the focus on OSBL turbulence, our approach also differs from theirs by a more tightly integrated coupling framework and the GPU acceleration of the embeded LES (see Section 2 for more details).

Simulating processes across scales is a major goal of MAPS-Ocean with its ability to regionally refine the resolution us-

ing unstructured horizontal grid. This allows regionally focused high-resolution simulations, e.g., in the coastal regions, in a





large background environment without being nested inside a coarse-resolution simulation. Yet the smallest scale MPAS-Ocean permits is essentially limited by the hydrostatic assumption. Therefore, coupling MPAS-Ocean with PALM is a natural extension of this multiscale modeling framework, which allows for extending multiscale simulations to a smaller scale in which non-hydrostatic dynamics becomes important.

### 1.4 Organization of This Paper

As an initial step towards a multiscale modeling framework of the OSBL turbulent mixing, this paper focuses on the implementation of the framework with some simplified test cases, leaving a more sophisticate demonstration of the applications of this framework in another study. The remainder of this paper is organized as follows. Section 2 describes the multiscale modeling framework, including the theory of coupling across scales, the approach to couple MPAS-Ocean with PALM and accelerating the PALM simulation using GPU. The coupled MPAS-Ocean and PALM are then tested in two idealized cases, a single column case and a mixed layer eddy case, in Section 3. The advantages and limitations of this approach, as well as some possible applications moving forward, are discussed in Section 4. This paper ends with a brief summary and main conclusions in Section 5.

## 2 Methods

### 2.1 Multiscale Modeling

As an illustration of the essential elements of coupling across scales, here we use a standard Reynolds decomposition to separate the dynamics into a large-scale and a small-scale. The large-scale and small-scale dynamics are then solved by different sets of equations. This is an implementation of the heterogeneous multiscale method (E et al., 2003, 2007). A broader review of multiscale modeling approaches in geophysical fluid applications can be found in Grooms and Julien (2018).

We begin with the non-hydrostatic Boussinesq equations for the momentum $\boldsymbol{u}$ and tracer $\theta$ that include both the large- and small-scale processes,

$$\partial_t \boldsymbol{u} = -\boldsymbol{u} \cdot \nabla \boldsymbol{u} - f\hat{\boldsymbol{z}} \times \boldsymbol{u} - \nabla p/\rho_0 + b\hat{\boldsymbol{z}} + \boldsymbol{d}, \tag{1}$$

$$\nabla \cdot \boldsymbol{u} = 0, \tag{2}$$

$$\partial_t \theta = -\boldsymbol{u} \cdot \nabla \theta + s, \tag{3}$$

where $f$ is the Coriolis parameter, $\hat{\boldsymbol{z}}$ the vertical unit vector, $\rho_0$ the reference density, $p$ the pressure, $\boldsymbol{d}$ the dissipation, $s$ the sources and sinks of tracer $\theta$. The buoyancy $b$ is related to the tracers by the equation of state $b = \mathcal{S}(\theta)$. For simplicity, only one tracer $\theta$ is shown here. But the equation of state generally depends on multiple tracers (such as the temperature and salinity of seawater), all satisfying the tracer equation in Eq. (3), and the pressure.



To isolate the large-scale process, consider a Reynolds decomposition,

$$\boldsymbol{u} = \overline{\boldsymbol{u}} + \boldsymbol{u}', \tag{4}$$

$$\theta = \overline{\theta} + \theta', \tag{5}$$

where $\overline{()}$ is the Reynolds average (in practice a spatial average over the horizontal grid cell in GCMs) and $()'$ is the fluctuation (unresolved in GCMs). Note that here multiscale is assumed only in the horizontal directions, though unlike previous studies, we are not assuming shared vertical coordinate between the large- and small-scales so that an interpolation in the vertical is necessary. Applying the Reynolds average to Eqs. (1)-(3), we get,

$$\partial_t \overline{\boldsymbol{u}} = -\overline{\boldsymbol{u}} \cdot \nabla \overline{\boldsymbol{u}} - f\hat{\boldsymbol{z}} \times \overline{\boldsymbol{u}} - \nabla \overline{p}/\rho_0 + \overline{b}\hat{\boldsymbol{z}} + \overline{\boldsymbol{d}} \underbrace{-\nabla_h \cdot \overline{\boldsymbol{u}'_h \boldsymbol{u}'} - \partial_z \overline{w'\boldsymbol{u}'}}_{\boldsymbol{F}^{\overline{\boldsymbol{u}}}_{\mathrm{SS}}}, \tag{6}$$

$$\nabla \cdot \overline{\boldsymbol{u}} = 0, \tag{7}$$

$$\partial_t \overline{\theta} = -\overline{\boldsymbol{u}} \cdot \nabla \overline{\theta} + \overline{s} \underbrace{-\nabla_h \cdot \overline{\boldsymbol{u}'_h \theta'} - \partial_z \overline{w'\theta'}}_{F^{\overline{\theta}}_{\mathrm{SS}}}, \tag{8}$$

where small-scale forcing terms are denoted as $\boldsymbol{F}^{\overline{\boldsymbol{u}}}_{\mathrm{SS}}$ and $F^{\overline{\theta}}_{\mathrm{SS}}$. The superscript indicates the variable to which the forcing term is applied. We have explicitly written out the horizontal and vertical components of the small-scale forcing terms as they are often treated separately in GCMs. Here $\nabla_h$ is the horizontal derivative, $\boldsymbol{u}_h$ and $w$ are the horizontal and vertical components of the velocity $\boldsymbol{u} = (\boldsymbol{u}_h, w)$.

The equations governing the small-scale fluctuations are derived by subtracting Eqs (6)-(8) from Eqs. (1)-(3),

$$\partial_t \boldsymbol{u}' = -\boldsymbol{u}' \cdot \nabla \boldsymbol{u}' - f\hat{\boldsymbol{z}} \times \boldsymbol{u}' - \nabla p'/\rho_0 + b'\hat{\boldsymbol{z}} + \boldsymbol{d}' \underbrace{+\nabla_h \cdot \overline{\boldsymbol{u}'_h \boldsymbol{u}'} + \partial_z \overline{w'\boldsymbol{u}'} - \overline{\boldsymbol{u}} \cdot \nabla \boldsymbol{u}' - \boldsymbol{u}' \cdot \nabla \overline{\boldsymbol{u}}}_{\boldsymbol{F}^{\boldsymbol{u}'}_{\mathrm{LS}}}, \tag{9}$$

$$\nabla \cdot \boldsymbol{u}' = 0, \tag{10}$$

$$\partial_t \theta' = -\boldsymbol{u}' \cdot \nabla \theta' + s' \underbrace{+\nabla_h \cdot \overline{\boldsymbol{u}'_h \theta'} + \partial_z \overline{w'\theta'} - \overline{\boldsymbol{u}} \cdot \nabla \theta' - \boldsymbol{u}' \cdot \nabla \overline{\theta}}_{F^{\theta'}_{\mathrm{LS}}}, \tag{11}$$

where large-scale forcing terms are denoted as $\boldsymbol{F}^{\boldsymbol{u}'}_{\mathrm{LS}}$ and $F^{\theta'}_{\mathrm{LS}}$.

So far we have been employing the standard Reynolds decomposition. A key step of multiscale modeling is to solve the large-scale (Eqs. (6)-(8)) and small-scale (Eqs. (9)-(11)) dynamics separately using different techniques. Different levels of coupling between scales are achieved by approximating the large- and small-scale forcing terms differently as detailed in the following.

The large-scale dynamics are usually solved by GCMs. At the scales that GCMs typically resolve, hydrostatic approximation (e.g., $\partial_t \overline{w} \ll \overline{b}$) is assumed so that the $\overline{w}$ equation reduces to the hydrostatic balance,

$$0 = -\partial_z \overline{p}/\rho_0 + \overline{b}. \tag{12}$$





In addition, the horizontal and vertical components of $\boldsymbol{F}_{\mathrm{SS}}^{\overline{\boldsymbol{u}}}$ and $F_{\mathrm{SS}}^{\overline{\theta}}$ are often parameterized separately due to the large aspect ratio of the grid cell (the horizontal extent of a cell over the vertical). Each component further consists of contributions from different processes. Traditional superparameterization approach estimates the vertical component of $\boldsymbol{F}_{\mathrm{SS}}^{\overline{\boldsymbol{u}}}$ and $F_{\mathrm{SS}}^{\overline{\theta}}$,

$$\boldsymbol{F}_{\mathrm{VSS}}^{\overline{\boldsymbol{u}}} = -\partial_z \overline{w'\boldsymbol{u}'}, \tag{13}$$

$$F_{\mathrm{VSS}}^{\overline{\theta}} = -\partial_z \overline{w'\theta'}, \tag{14}$$

from a process-resolving model that solves for $\boldsymbol{u}'$ and $\theta'$. It replaces the conventional vertical mixing parameterization, but essentially retains the same scale separation and conditional equilibration assumptions. The horizontal component of $\boldsymbol{F}_{\mathrm{SS}}^{\overline{\boldsymbol{u}}}$ and $F_{\mathrm{SS}}^{\overline{\theta}}$ is assumed to result from large-scale processes that are decoupled to the small-scale processes solved by the embedded process-resolving model with a small horizontally homogeneous domain. Such large-scale processes are represented by other parameterizations in GCMs. The *point approximation* approach proposed by Grooms and Majda (2013) allows the horizontal

component of $\boldsymbol{F}_{\mathrm{SS}}^{\overline{\boldsymbol{u}}}$ and $F_{\mathrm{SS}}^{\overline{\theta}}$ from the embedded small-scale dynamics in additional to the vertical. However, the relevance of the horizontal components depends on the spatial and temporal scales of the dynamics governing the $\boldsymbol{u}'$ and $\theta'$ fields. Scale separation of the dynamics among more than two scales may be necessary, as commonly assumed by separate parameterizations of different processes in GCMs, such as mesoscale eddies, which contributes mostly to the horizontal fluxes, and submesoscale eddies and boundary layer turbulence, both of which contribute mostly to the vertical fluxes. For the purpose here to represent

the ocean surface turbulent mixing in GCMs, we only explicitly consider the vertical component of $\boldsymbol{F}_{\mathrm{SS}}^{\overline{\boldsymbol{u}}}$ and $F_{\mathrm{SS}}^{\overline{\theta}}$ in Eqs. (6) and (8), while leaving the horizontal component being parameterized.

   For the small-scale dynamics, a local embedded domain is commonly used in both the traditional superparameterization approach and the point approximation approach. The point approximation approach solves Eqs. (9)-(11) for $\boldsymbol{u}'$ and $\theta'$ on a local embedded horizontally periodic domain in which the mean fields $\overline{\boldsymbol{u}}$ and $\overline{\theta}$ are constant and have constant derivatives

between the coupling steps with the large-scale equations. In the embedded domain, it is necessary to differentiate the horizontal derivatives of large-scale and small-scale fields. Denoting the large-scale lateral gradient as $\overline{\nabla}_h$ and small-scale lateral gradient as $\nabla'_h$, the large-scale forcing terms in Eqs. (9) and (11) can be written as,

$$\boldsymbol{F}_{\mathrm{LS}}^{\boldsymbol{u}'} = \overline{\nabla}_h \cdot \overline{\boldsymbol{u}'_h \boldsymbol{u}'} + \partial_z \overline{w' \boldsymbol{u}'} - \overline{\boldsymbol{u}}_h \cdot \nabla'_h \boldsymbol{u}' - \boldsymbol{u}'_h \cdot \overline{\nabla}_h \overline{\boldsymbol{u}} - w' \partial_z \overline{\boldsymbol{u}}, \tag{15}$$

$$F_{\mathrm{LS}}^{\theta'} = \overline{\nabla}_h \cdot \overline{\boldsymbol{u}'_h \theta'} + \partial_z \overline{w' \theta'} - \overline{\boldsymbol{u}}_h \cdot \nabla'_h \theta' - \boldsymbol{u}'_h \cdot \overline{\nabla}_h \overline{\theta} - w' \partial_z \overline{\theta}, \tag{16}$$

where $\overline{w} = 0$ is assumed in the embedded domain.

   Alternatively, one can solve the small-scale dynamics by solving for the total momentum $\boldsymbol{u}$ and tracer $\theta$ in the embedded domain, and enforce

$$\langle \boldsymbol{u}_h \rangle = \overline{\boldsymbol{u}}_h, \tag{17}$$

$$\langle \theta \rangle = \overline{\theta}, \tag{18}$$

locally at each coupling stage where $\langle \rangle$ is the average over the embedded domain. Writing both Eqs. (6)-(8) and Eqs. (9)-(11) on the local embedded domain and adding them together, noting that $\nabla'_h \overline{\phi} = 0$ where $\phi$ represents either momentum or tracers,





we get,

$$\partial_t \boldsymbol{u} = -\boldsymbol{u} \cdot \nabla' \boldsymbol{u} - f \hat{\boldsymbol{z}} \times \boldsymbol{u} - \nabla' p'/\rho_0 + b' \hat{\boldsymbol{z}} + \boldsymbol{d} \underbrace{-\overline{\nabla}_h \overline{p}/\rho_0 - \boldsymbol{u}_h \cdot \overline{\nabla}_h \overline{\boldsymbol{u}}}_{\boldsymbol{F}_{\mathrm{LS}}^{\boldsymbol{u}}}, \tag{19}$$

$$\nabla' \cdot \boldsymbol{u} = 0, \tag{20}$$

$$\partial_t \theta = -\boldsymbol{u} \cdot \nabla' \theta + s \underbrace{-\boldsymbol{u}_h \cdot \overline{\nabla}_h \overline{\theta}}_{F_{\mathrm{LS}}^{\theta}}, \tag{21}$$

where $\nabla' = (\nabla'_h, \partial_z)$ and the hydrostatic balance in Eq. (12) is used. Ignoring the large-scale lateral gradient terms $\boldsymbol{F}_{\mathrm{LS}}^{\boldsymbol{u}}$ and $F_{\mathrm{LS}}^{\theta}$, Eqs. (19)-(21) reduce to the equations in the traditional superparameterization approach. The constraints (17) and (18) are enforced by either applying a nudging term in Eqs. (19) and (21) (e.g., Khairoutdinov et al., 2005) or mapping from $\overline{\boldsymbol{u}}_h$ and $\overline{\theta}$ to $\boldsymbol{u}_h$ and $\theta$ to keep them consistent at the beginning of each coupling step (e.g., Campin et al., 2011). In the

superparameterization approach, the embedded domain does not necessarily fill the large-scale GCM grid cell over which the spatial average replaces the Reynolds average. However, to allow the large-scale forcing terms $\boldsymbol{F}_{\mathrm{LS}}^{\boldsymbol{u}}$ and $F_{\mathrm{LS}}^{\theta}$ in Eqs. (19) and (21), appropriate estimates of the lateral gradients have to be made. The relevance of these terms therefore depends on the ratio of the large- and small-scales of interest. See Section 4 for more discussion on this issue.

## 2.2 Coupling MPAS-Ocean with PALM

The Model for Prediction Across Scales-Ocean (MPAS-Ocean, Ringler et al., 2013; Petersen et al., 2018) is the ocean component of the U.S. Department of Energy's Earth system model, the Energy Exascale Earth System Model (E3SM, Golaz et al., 2019). It solves the hydrostatic, incompressible, and Boussinesq primitive equations on an unstructured-mesh using finite volume discretization. Model domains may be spherical with realistic bottom topography to simulate the Earth's oceans, or Cartesian for idealized experiments. The PArallelized Large eddy simulation Model (PALM, Raasch and Schröter, 2001; Maronga

et al., 2015) is a turbulence-resolving LES model to simulate turbulent flows in the atmospheric and oceanic boundary layers. It solves the non-hydrostatic, incompressible and spatially filtered Navier–Stokes equations with the Boussinesq approximation on Cartesian grid using finite difference discretization. It has been widely used to simulate a variety of processes in planetary boundary layers (see Maronga et al., 2015, and the references therein). Both MPAS-Ocean and PALM are under extensive development (see the latest versions at github.com/MPAS-Dev/MPAS-Model/releases and palm.muk.uni-hannover.de/trac). The

coupled MPAS-Ocean and PALM presented here is based on MPAS-Ocean version 6.3 and PALM version 5.0.

To couple MPAS-Ocean with PALM, an interface between the two models to exchange information was developed. In particular, the PALM main driver was modularized by wrapping up all the necessary subroutines into three separate steps: initialization, time-stepping and finalization. PALM can now be compiled in either the standalone mode or the modular mode, the latter of which can be easily used in other GCMs too. MPAS-Ocean is coded in a modular way such that different parame-

terizations can be easily changed depending on the input namelist. The PALM module is therefore used in MPAS-Ocean as an additional option for vertical mixing, replacing the ocean surface vertical mixing schemes such as the K-Profile Parameterization (KPP, Large et al., 1994; Van Roekel et al., 2018) where needed.





As a first step towards a multiscale modeling framework outlined in Section 2.1, MPAS-Ocean is coupled with PALM using a simple approach similar to the traditional superparameterization, with a few important exceptions for enhanced flexibility. MPAS-Ocean solves the large-scale momentum and tracer equations (6)-(8) on a coarse grid. PALM solves the total momentum and tracer equations (19)-(21) on a fine grid, embedded in a grid cell of the coarse MPAS-Ocean grid. The small-scale to large-scale coupling is achieved by estimating the vertical convergence of small-scale turbulent fluxes of (13) and (14) from the domain averaged fluxes in PALM,

$$\boldsymbol{F}_{\mathrm{VSS}}^{\overline{\boldsymbol{u}}} = -\partial_z \langle w' \boldsymbol{u}' \rangle, \tag{22}$$

$$F_{\mathrm{VSS}}^{\overline{\theta}} = -\partial_z \langle w'\theta' \rangle, \tag{23}$$

and passing them to MPAS-Ocean at each MPAS-Ocean time step. Here the fluxes are also averaged over an MPAS-Ocean time step. See Fig. 1 for a schematic diagram of the coupling in time. Large-scale to small-scale coupling is done by enforcing Eqs. (17) and (18) using nudging terms in Eqs. (19) and (21),

$$\boldsymbol{F}_{\mathrm{LS}}^{\boldsymbol{u}} = \frac{\overline{\boldsymbol{u}}_h - \langle \boldsymbol{u}_h \rangle}{\tau_{\mathrm{LS}}^{\boldsymbol{u}}}, \tag{24}$$

$$F_{\mathrm{LS}}^{\theta} = \frac{\overline{\theta} - \langle \theta \rangle}{\tau_{\mathrm{LS}}^{\theta}}, \tag{25}$$

where $\tau_{\mathrm{LS}}^{\boldsymbol{u}}$ and $\tau_{\mathrm{LS}}^{\theta}$ are the nudging time scales for momentum and tracers. Both nudging terms are estimated at the beginning of each coupling step and held constant until the next coupling step. Each of the coupling terms can be switched on and off separately so that various levels of coupling can be achieved for sensitivity tests. A reasonable choice for $\tau_{\mathrm{LS}}^{\theta}$ is the time step of MPAS-Ocean (the coupling period here), which is consistent with Campin et al. (2011). Campin et al. (2011) also suggest a nudging time scale for momentum (in their case nudging the direction) shorter than the flow adjustment time (e.g., inertial period) but longer than the coarse-resolution model's time step to prevent sudden changes of orientation due to numerical noise. Here we tested different values of $\tau_{\mathrm{LS}}^{\boldsymbol{u}}$ from 30 minutes to 5 hours. A relatively long $\tau_{\mathrm{LS}}^{\boldsymbol{u}}$ of a few hours was necessary in our case to alleviate the spurious influence of neighboring cells when PALM is only running in selected MPAS-Ocean grid cells – a feature of our approach that will be introduced later in this section. We therefore defer the details of this issue to later in this section and in Section 3. While part of the influences of lateral gradient in MPAS-Ocean are felt by PALM through the nudging terms, their effects on the small-scale dynamics are ignored here for simplicity. The effects of nonzero large-scale lateral gradient on the small-scale dynamics are discussed in Section 4 and will be explored in a future study.

In contrast to many previous versions of superparameterization, here we do not assume that the coarse-resolution fields and fine-resolution fields are on the same vertical grid. Therefore, a remapping step between coarse vertical grid and fine vertical grid is necessary. Here we use the piecewise quartic method described in White and Adcroft (2008), realized by a high-order Piecewise Polynomial Reconstruction library (PPR, github.com/dengwirda/PPR). Although this remapping method is conservative, monotonic and highly accurate, loss of information is unavoidable, especially when remapping from high-resolution to low-resolution. When remapping the PALM fields to the MPAS-Ocean grid, the averaged effect of the high-resolution PALM is applied to the low-resolution MPAS-Ocean, which is what we want. But when applying the large-scale





forcing from low-resolution MPAS-Ocean to the high-resolution PALM fields, a simple nudging to the MPAS-Ocean fields
will cause loss of information, e.g., near the base of the mixed layer where the gradients are strong. To reduce such loss of
information due to remapping, both the large-scale and small-scale forcings are evaluated on the original vertical grid and then
remapped to the targeted vertical grid. In this way, the relatively sharp vertical gradients in PALM fields that are not resolved
in MAPS-Ocean are preserved.

Instead of initializing a PALM instance at each coupling step and running it to quasi-equilibrium, we initialize the LES at
the beginning of the MPAS-Ocean simulation (by calling the initialization subroutine) and let it run throughout the simulation
(by repeatedly calling the time-stepping subroutine to step forward). See Fig. 1 for an illustration of MAPS-Ocean and PALM
running in parallel. This is intrinsically different from using LES as a replacement of the conventional parameterizations (thus
the term superparameterization), in which an equilibrium of turbulence to the changing external forcing is often assumed.

The equilibrium assumption may be valid if the time step of the coarse-resolution model is long enough, within which the
turbulence can adjust to the changes in the external forcing. However, as the time step of the ocean GCMs become shorter and
shorter (half an hour for a common global MPAS-Ocean simulation but much shorter for regional simulations), the equilibrium
assumption may no longer be valid. The coupling approach here allows for disequilibrium of turbulence when the forcing
conditions change rapidly.

As in KPP, PALM is running at the center of an MPAS-Ocean cell. A mask variable is introduced to allow PALM to run
only on selected MPAS-Ocean grid cells. KPP is used for other grid cells following Van Roekel et al. (2018). Fig. 2 shows
a schematic diagram of this flexible layout of coupling MPAS-Ocean with PALM in space. In this example, PALM is only
running on cell $c_6$ whereas all other cells use KPP. The tracer fluxes from PALM are directly used to update the tracer equations
at $c_6$ via Eqs. (8) and (23). The momentum equation Eq. (6) is solved in MPAS-Ocean by solving the normal velocity equation

at all the edges (Ringler et al., 2010). Therefore, the momentum fluxes from both PALM at $c_6$ and KPP at $c_{11}$ contribute to the
normal velocity at edge $e_{6|11}$. In practice, half of the momentum fluxes from PALM running at the center of a certain cell are
applied to its edges. This is consistent with the present configuration of KPP in MPAS-Ocean, in which the vertical viscosity
at the edges are the average of the values at its two neighboring cells. Since the KPP vertical viscosity at the center of a cell
is set to zero whenever PALM is running on that cell, this approach avoids double counting of the momentum fluxes at the

edges and allows for smooth transition from PALM cells to KPP cells, regardless of the layout of PALM cells. However, this
also means that the PALM cells are essentially coupled with the neighboring KPP cells. When the momentum of MPAS-Ocean
and PALM are tightly coupled, e.g., through a nudging term with a short time scale, the solution of neighboring KPP cells will
influence the solution in PALM, which is undesirable. Using a relatively longer nudging time scale of a few hours alleviates
this problem. We return to this issue in Section 3 using an idealized diurnal heating and cooling case in the single column

MPAS-Ocean setup.

     Running PALM on only selected MPAS-Ocean grid cells may cause a load imbalance, as the grid columns of MPAS-Ocean
with PALM running will be much slower than those without. This problem can be alleviated by running PALM on GPU (see the
next section) and carefully designing the mesh decomposition to balance the CPU jobs and GPU jobs on each computational
node.





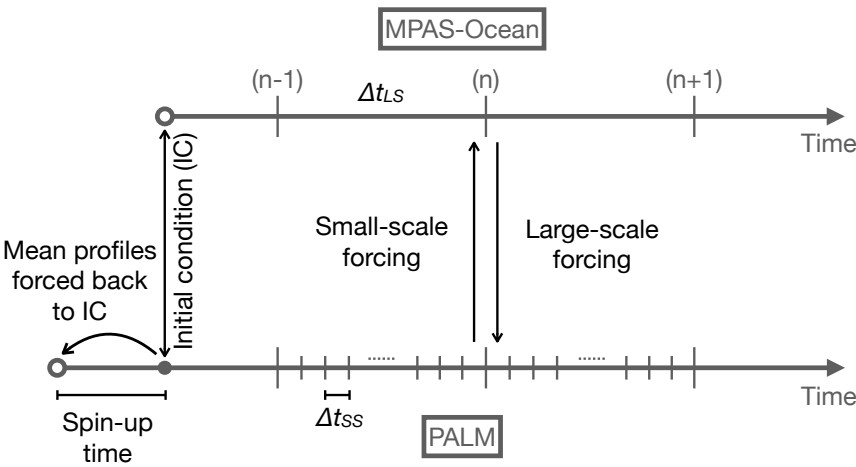

**Figure 1.** Schematic diagram of coupling MPAS-Ocean with PALM in time. PALM and MPAS-Ocean are running in parallel with time steps $\Delta t_{\mathrm{LS}}$ and $\Delta t_{\mathrm{SS}}$, respectively, and exchange information at every $\Delta t_{\mathrm{LS}}$. PALM estimates the small-scale vertical momentum and tracer fluxes due to turbulent mixing, whose vertical divergence in Eqs. (22) and (23) are used in MAPS-Ocean. MPAS-Ocean provides PALM the surface forcing and the large-scale forcing, which are nudging terms following Eqs. (24) and (25) here, but can be extended to include the lateral gradients in Eqs. (19) and (21). PALM also includes a spin-up phase after which the turbulent fluctuations are preserved while the mean profiles are forced back to the initial conditions following Eq. (27). See the text for more details.

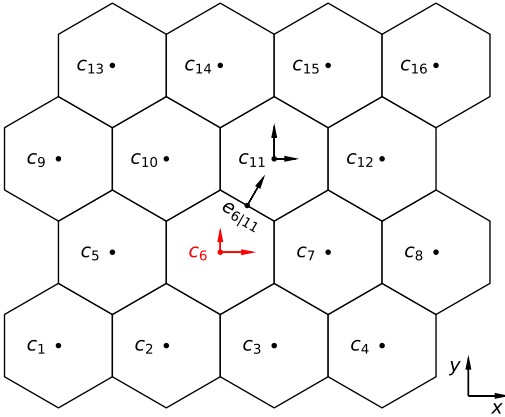

**Figure 2.** Schematic diagram of coupling MPAS-Ocean with PALM with PALM running only on selected cells. In this example, PALM is running at the center of cell $c_6$ (in red) whereas KPP is used in all other cells (in black). The normal velocity at the edge $e_{6|11}$ is therefore affected by both PALM at $c_6$ and KPP at $c_{11}$. See the text for more details.





### 2.3 PALM on GPU

The growing computational power of modern supercomputers, especially with the acceleration using GPUs, allows us to explore the possibility of deploying the embedded PALM on GPU. Since there is no need of communications among different instances of the embedded PALM, they may run efficiently on GPU in parallel in massive scale. This section describes our effort to port a customized version of PALM on GPU.

Significant modification and reduction of the PALM code (version 5.0) were conducted before porting it on GPU. In particular, all the special treatment of complex topography and surface types were removed, along with the atmospheric variables, all of which are irrelevant to our OSBL application here. We also removed the option to use different numerical schemes and hard-coded it to use the 5th and 6th order advection scheme of Wicker and Skamarock (2002) and 3rd order Runge-Kutta time-stepping scheme. We used an external fast Fourier Transform library (FFTW) for the pressure solver (an option in PALM) in the CPU version for benchmarking. These steps significantly reduced the amount of effort required to port PALM on GPU.

This customized version of PALM was ported on GPU using OpenACC directive-based parallel programming model (openacc.org) and the NVIDIA CUDA Fast Fourier Transform library (cuFFT, developer.nvidia.com/cufft). Porting PALM involved two iterative steps: (1) parallelizing the loops and (2) optimizing data locality. In the first step we wrapped all the loops in the time integration subroutine with the OpenACC "parallel" directive, which allows for automatic parallelization of loops. Some loops, especially in the tridiagonal solver and the turbulence closure subroutines, were restructured to ensure independency. Lower degree of parallelism was employed for loops that cannot be easily restructured to remove the dependency between iterations. This step generally slows down the code because of the large amount of data exchanges between the CPU and the GPU occurs. Therefore, optimizing data locality is required. This was done by enlarging the data regions of the parallelized loops on GPU. Then we moved on to parallelize another segment of the code and iterated between the two steps.

In the process of porting the code, some loops are executed on the CPU and some on the GPU. So extreme caution is required to make sure the data are synchronized between the CPU and the GPU when necessary. Eventually, the data region on GPU is large enough to cover the entire time integration subroutine. This reduces the data exchange between the CPU and the GPU to mostly at the beginning and the end of the time integration, which will occur once per MPAS-Ocean time step. As most of the subroutines during the time integration are ported on GPU, values of the variables are updated on the CPU rather infrequently only when it is necessary. This significantly reduces the time-consuming data exchange between the CPU and the GPU, which leads to overall acceleration of the code. The cuFFT library was used to accelerate the pressure solver in PALM, which uses fast Fourier transform to solve the Poisson's equation.

The speedup of porting PALM on GPU was benchmarked by running the standalone PALM with and without GPU on two machines: (1) a Linux workstation with an Intel Xeon Silver 4112 CPU @ 2.60GHz and an NVIDIA Quadro RTX 4000 GPU; and (2) the High Performance Computing system at Oak Ridge National Laboratory (Summit) with 2 IBM POWER9 CPUs and 6 NVIDIA Tesla V100 GPUs on each node. For the benchmarking on Summit, only 1 CPU and 1 GPU were used. Fig. 3 shows the speedup in total run time and the three most time-consuming subroutines. The speedup factor is defined as the ratio of the run time on 1 CPU divided by the run time on 1 CPU and 1 GPU. Overall a 10-16 times speedup was achieved by porting PALM



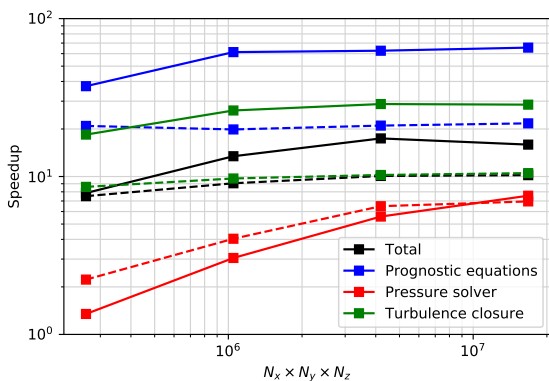

**Figure 3.** Speedup factor as a function of problem size ($N_x \times N_y \times N_z$). The speedup factor is defined here as the ratio of the run time on 1 CPU divided by the run time on 1 CPU + 1 GPU. The black line shows the speedup in total time and colored lines show the speedup in the three most time-consuming subroutines. Solid lines show the speedup on Summit and dashed lines on the Linux workstation.

on GPU, especially for relatively large problem sizes for which the full capability of the GPU can be used. For subroutines that

have many independent loops, such as the prognostic equations, the speedup can be larger than a factor of 65. However, the overall speedup was strongly limited by the pressure solver in which the Fourier transform restricts the degree of parallelism, especially for smaller problems. Note that this speedup is achieved by a straightforward application of the OpenACC parallel directives to parallelize the loops and copying the data to GPU at the beginning of the time integration. Further speedup is possible by improving the memory management, and fine-tuning the parallelization of loops and data locality to better suit the

GPU capability, which are beyond the scope of this study.

## 3 Results

Here we validate the coupled MPAS-Ocean and PALM using two idealized test cases. The first and simplest is in a single column configuration, which was used throughout the development. The goal of this configuration is a test of one-way coupling from small-scale to large-scale dynamics in Eqs. (6) and (8) by estimating the small-scale forcing terms in Eqs. (22) and (23)

from PALM. This also allows a test of the remapping between the vertical grids of MPAS-Ocean and PALM.

The second test case is a simulation of mixed layer eddies with PALM running on multiple MPAS-Ocean grid cells. The goal of this test case is twofold. First, this case allows us to test the capability of running multiple PALM instances within MPAS-Ocean and distributing the computation on multiple GPUs. Second, this case provides a way to test the two-way coupling between MPAS-Ocean and PALM under more complex and realistic conditions. In particular, the development of mixed layer

eddies in this case introduces spatial heterogeneity and large-scale forcing from advection and lateral mixing, though baroclinic instability in PALM is excluded as a result of missing the lateral gradients.





### 3.1 Single Column Test

MPAS-Ocean allows for a minimum of 16 columns in a "single column" mode, in which the 16 columns are forced by the same surface forcing and are essentially identical to each other due to the lack of lateral processes. Therefore, ideally running PALM

in the single column MPAS-Ocean is essentially the same as running the standalone PALM as there should be no large-scale forcing terms. In practice, this is true only if PALM is running in all 16 columns. If PALM is only running on one of these columns while KPP is used in other columns, as illustrated in Fig. 2, different solutions between PALM and KPP under the same forcing conditions will result in some spurious large-scale forcing to PALM. This is especially the case for the momentum since the momentum is solved on cell edges in MAPS-Ocean, i.e., affected by the momentum fluxes from both the PALM cell

and the KPP cells. This 16-cell "single column" MPAS-Ocean configuration allows us to assess the impact of this issue and explore possible remedies.

The coupled MPAS-Ocean and PALM is tested in this single column configuration under different idealized forcing scenarios using various combinations of constant wind stress, constant surface cooling and diurnal solar heating. Here we only present one case in detail as an example. In this case the simulation is initialized from a stable stratification with surface temperature of

$20\,^\circ$C and a constant vertical gradient of $0.01\,^\circ$C/m. Salinity is constant at 35 psu to make sure the coupling does not introduce spurious tendencies. We run the MPAS-Ocean simulation from rest for two days with a time step of 30 min. The depth of the simulation domain is 102.4 m, divided into 80 layers with constant layer thickness of 1.28 m. For some configurations we also repeated the same simulation with a coarse vertical resolution of 5.12 m to test the sensitivity to vertical remapping.

The surface forcing is an idealized diurnal heating and cooling with constant wind stress of 0.1 N/m$^2$. An idealized diurnal

cycle of the solar radiation $F_s$ is used,

$$F_s(t) = F_0 \max\left\{ \cos\left[ 2\pi \left( \frac{t}{86400} - 0.5 \right) \right], 0 \right\}, \tag{26}$$

where $F_0 = 500$ W/m$^2$ is the maximum solar radiation at noon and $t$ is the time of a day in seconds. Absorption of solar radiation in the water column is computed using the two-band approximation of Paulson and Simpson (1977) assuming Jerlov water type IB. A constant surface cooling of $-159.15$ W/m$^2$ is applied at the surface, balancing the solar heating when

integrated over a day. Coriolis parameter is set to $1.028\times10^{-4}$ s$^{-1}$, equivalent to the value at a latitude of 45°N. This test case yields an inertial oscillation in the velocities with a period of about 17 hours in addition to the diurnal heating and cooling.

The coupled MPAS-Ocean and PALM is tested by running PALM only at the sixth cell $c_6$ (see Fig. 2). All other 15 cells use KPP for the vertical mixing with a typical configuration (Van Roekel et al., 2018). PALM runs on a 160 m×160 m×80 m domain with 128×128×80 grid cells, yielding a resolution of $[dx, dy, dz] = [1.25, 1.25, 1]$ m. This configuration of PALM

is enough to give statistically robust mean vertical profiles of the fluxes that are exchanged with MPAS-Ocean. PALM starts with some small random perturbations on the temperature and velocity fields (between depths of around 4 m and 28 m) during the first 150 s, and then runs for one hour to allow the turbulence to develop before the initial step of MPAS-Ocean. Then the mean fields of PALM are forced back to the initial profiles of MPAS-Ocean described above, while keeping the turbulent





perturbations developed in the PALM initial step,

$$\phi^*(x,y,z) = \phi(x,y,z) - \langle\phi\rangle(z) + \overline{\phi}(z), \tag{27}$$


where $\phi^*$ is the updated PALM field, $\phi$ the original, $\langle\phi\rangle$ the horizontal average over the PALM domain, and $\overline{\phi}$ the initial condition from MPAS-Ocean (see Fig. 1).

We first compare the solutions of this test case in standalone PALM and KPP in Fig. 4. It is clearly seen that in this test case KPP gives quite different solutions than the standalone PALM. In particular, KPP yields slightly stronger vertical mixing

indicated by the deeper boundary layer by a few meters. As a result, the simulated temperature is cooler throughout most part and warmer near the base of the boundary layer (panel d), and the maximum sea surface temperature (SST) during a diurnal cycle is about 0.04 °C cooler (panel a) in KPP than in PALM. The simulated velocities are clearly affected by the diurnal heating during different phases of the inertial oscillation in PALM, but not in KPP.

For the coupled MPAS-Ocean and PALM, with the small-scale forcing Eqs. (22) and (23) from PALM applied to all the

16 cells, we expect the results being identical to the standalone PALM, except perhaps some small differences due to the differences in time stepping. This is indeed the case, as shown by comparing the blue line and black line in Fig. 5. Here the profiles in Fig. 5 are taken at the time of SST maximum indicated by the dotted line in Fig. 4.

Since the small-scale forcing of tracers in Eq. (23) is directly applied to the tracer equations at the MPAS-Ocean cell centers, the coupling of tracers (here temperature) is straightforward even if PALM is running on one cell $c_6$ (Fig. 2) as there is no

direct impact from neighboring cells. This can be seen from the agreement of the dark red line and the black line in Fig. 5a, in which $\tau_{LS}^{\boldsymbol{u}} = \infty$ or equivalently $\boldsymbol{F}_{LS}^{\boldsymbol{u}} = 0$ following Eq. (24) effectively cuts off the influence of neighboring KPP cells on the momentum equations in PALM. However, as the momentum at the center of the PALM cell ($c_6$) is directly coupled with the neighboring KPP cells (by solving only the normal velocity at edges), we see some differences as the coupling of the momentum gets tighter with shorter nudging time scale for the momentum (see the transition from dark red to light red in

Fig. 5). In particular, although the momentum profiles appear to be slightly closer to the standalone PALM as the nudging is stronger, the temperature starts to be affected by this strong coupling with neighboring cells. The surface temperature (upper 10 meters) gets warmer due to the diurnal heating and the layer immediately below gets slightly cooler as compared to the standalone PALM. This is probably an artifact due to the mismatching tendencies of warmer surface temperature from PALM and weaker velocity shear from KPP, making the surface layer in the embedded PALM more stable. The momentum near the

base of the boundary layer is also strongly influence by neighboring cells running KPP. Similar behavior is also seen in a pure shear driven entrainment case (not shown).

The similar results between the coarse vertical resolution (gray line) and the fine vertical resolution (red line) in Fig. 5a suggest that embedding a high-fidelity LES is able to improve the temperature distribution of a coarse vertical resolution calling model, and the error of vertical remapping with the piecewise quartic method (White and Adcroft, 2008) is minimal.

However, embedding a fine vertical resolution KPP doesn't help, indicated by the similar results of black and gray dashed lines in Fig. 5a (in this case the coarse resolution KPP appears to perform better near the boundary layer base, see also Van Roekel et al. (2018)). Some improvements over KPP in the simulated momentum are also seen in both the fine vertical resolution



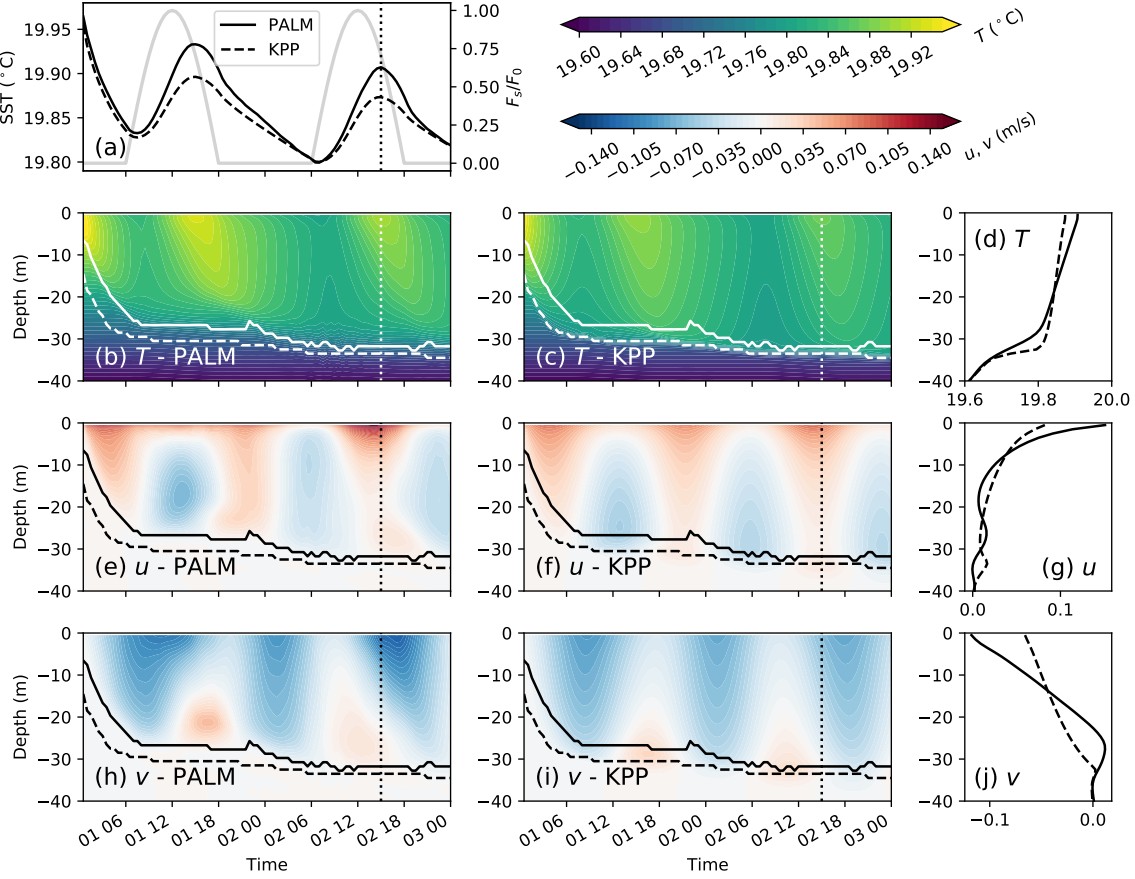

**Figure 4.** A comparison between KPP and standalone PALM simulations in the diurnal heating case. Panel (a) shows the simulated sea surface temperature (SST, °C) in standalone PALM (solid line in black) and KPP (dashed line). For reference the diurnal factor ($F_s/F_0$) in Eq. (26) is also shown (gray line, vertical axis on the right). Panels (b,c), (e,f), and (h,i) show the simulated temperature ($T$, °C) and velocities ($u$, $v$, m/s) in $x$- and $y$-directions, respectively, in PALM and KPP. Solid and dashed lines mark the boundary layer base in PALM and KPP, respectively, defined as the depth where the stratification reaches its maximum. The dotted line in these panels marks the time when the profiles in PALM and KPP are compared in (d), (g) and (j), respectively.



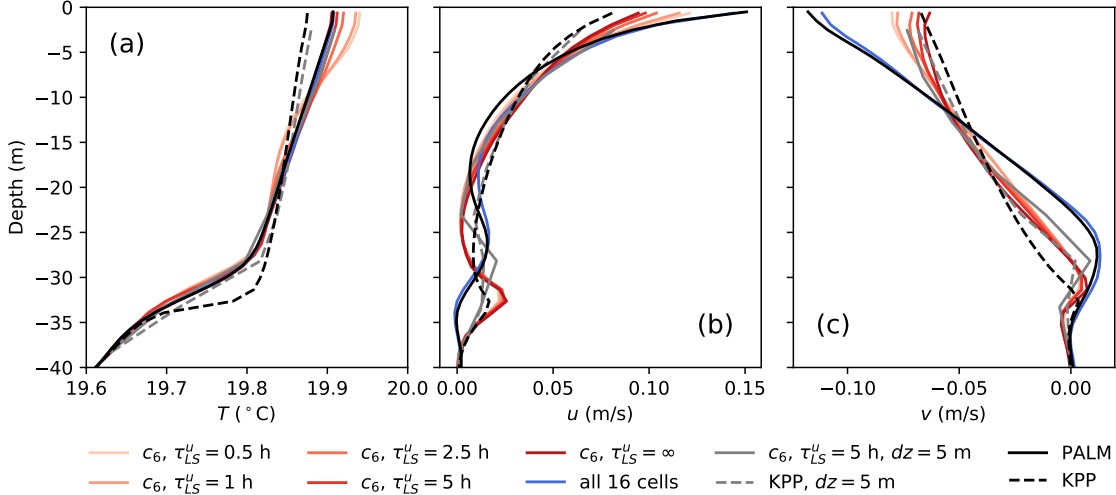

**Figure 5.** A comparison of the temperature (a), and zonal velocity (b) and meridional velocity (c) profiles with different configurations and coupling strategies. The profiles are taken at the SST maximum (about 3 hours after the solar radiation maximum) during the second diurnal cycle of the diurnal heating cases (dotted line in Fig. 4). On top of the profiles in the standalone PALM (black solid) and KPP (black dashed) from panels (d), (g) and (j) of Fig. 4, we are overlaying the profiles from a set of sensitivity tests with different configurations and coupling strategies. Light to dark red lines show the results of simulations with different momentum nudging time scale from $\tau_{\mathrm{LS}}^{\boldsymbol{u}} = 0.5$ h to zero momentum nudging ($\tau_{\mathrm{LS}}^{\boldsymbol{u}} = \infty$). Blue line shows the results of a simulation in which the tendencies of PALM are applied to all 16 cells, versus only cell $c_6$ in other cases. Gray solid and dashed lines show the same as the $c_6$, $\tau_{\mathrm{LS}}^{\boldsymbol{u}} = 5$ h case and the KPP case, respectively, but with MPAS-Ocean running on a coarse resolution of $dz = 5.12$ m. In all cases (when applicable) the nudging time scale for tracers is $\tau_{\mathrm{LS}}^{\theta} = 0.5$ h.

case (red solid lines in Fig. 5b and c) and the coarse vertical resolution case (gray solid lines in Fig. 5b and c). But since the momentum at the cell center is reconstructed from the normal velocities at all edges of a cell, which are strongly influenced by

the neighboring KPP cells, it's profiles resemble more the momentum profiles in KPP than that in PALM.

Spatial influence of running PALM on the cell center of a single cell in the 16-cell single column configuration is shown in Fig. 6. The difference between all the 15 KPP cells and the PALM cell (marked by the plus sign) for temperature is roughly uniform, except the two adjacent cells on the left and right, and the cell at the upper right (panel a, note that the domain is doubly periodic), likely due to the feedback from changes in the velocity in those cells (panels b and c). The changes in zonal

and meridional velocities are results of the interpolation of small-scale forcing terms from cell center to cell edges, and the reconstruction of the zonal and meridional velocities from normal velocities at edges.

Note that running PALM on the cell edges of MPAS-Ocean (e.g., on $e_{6|11}$ in Fig. 2) wouldn't help to eliminate this issue in the coupling of the momentum. At first thought it might appear promising as in that case the normal velocity at the edge can be directly used to update the momentum in PALM. However, since the tangential velocity in MPAS-Ocean (Ringler et al., 2010)

is diagnosed from the normal velocity at all the edges of the two neighboring cells (e.g., $c_6$ and $c_{11}$) of an edge (e.g., $e_{6|11}$),

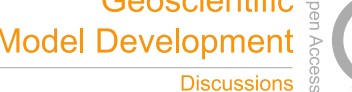

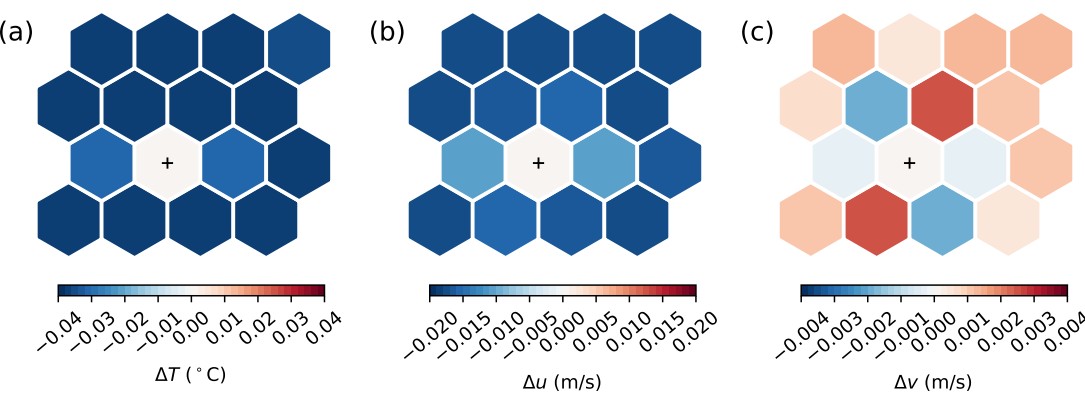

**Figure 6.** Distribution of the differences in temperature (a), zonal velocity (b) and meridional velocity (c) at the surface between the KPP cells and the PALM cell (marked by "+") in the single column test. Here the nudging time scale for the momentum and tracers are $\tau_{\mathrm{LS}}^{u} = 5$ h and $\tau_{\mathrm{LS}}^{\theta} = 0.5$ h.

the influence of neighboring KPP cells is still there. We therefore choose the present configuration of running PALM on the MPAS-Ocean cell centers and use a relatively long (5 hours) nudging time scale for the momentum.

### 3.2 Mixed Layer Eddy Test

The setup of the mixed layer eddy test case is guided by similar simulations in both GCMs (Fox-Kemper et al., 2008) and LES
models (Hamlington et al., 2014). However, since our focus is not on the mixed layer eddy itself, we are not comparing the simulation results here with those in the literature. Instead we focus on the coupling between MPAS-Ocean and PALM in the presence of some mixed layer eddies.

Here we run MPAS-Ocean with a two-front, or warm filament, setup on a domain of 72 km × 62.4 km × 150 m with 14400 cells and 50 vertical levels, corresponding to horizontal (distance between cell centers) and vertical grid sizes of 600 m and
3 m, respectively. The initial temperature field $T_0$ is uniformly distributed in the $x$-direction with two fronts in the $y$-direction,

$$T_0(y,z) = T_b + \frac{N^2}{\alpha_T g}(z+H) + \frac{L_f M_f^2}{2\alpha_T g}\left\{1 + \tanh\left[\frac{2(y-y_1)}{L_f}\right] - \tanh\left[\frac{2(y-y_2)}{L_f}\right]\right\}, \tag{28}$$

$$N^2 = \begin{cases} N_{\mathrm{ml}}^2, & \text{if } z > -H, \\ N_{\mathrm{int}}^2, & \text{if } z \leq -H, \end{cases} \tag{29}$$

where $T_b = 16$ °C is the temperature at the base of the mixed layer, $H = 50$ m the mixed layer depth, $N_{\mathrm{ml}}^2 = 1.96 \times 10^{-7}$ s$^{-2}$ the stratification within the mixed layer and $N_{\mathrm{int}}^2 = 1.96 \times 10^{-5}$ s$^{-2}$ below, corresponding to $\partial_z T = 10^{-4}$ °C/m and $\partial_z T =$
$10^{-2}$ °C/m, respectively, using a linear equation of state with the thermal expansion coefficient $\alpha_T = 2 \times 10^{-4}$ °C$^{-1}$ and the gravity $g = 9.81$ m/s$^2$. The front width is $L_f = 10$ km, with the horizontal buoyancy gradient $M_f^2 = 3.92 \times 10^{-8}$ s$^{-2}$





($\partial_y T = 2 \times 10^{-5}$ °C/m). The initial front locations are $y_1 = 15.6$ km and $y_2 = 46.8$ km. The salinity is constant at 35 psu. The Coriolis parameter is $f = 10^{-4}$ s$^{-1}$. These parameters corresponds to a Richardson number $Ri = N_{\mathrm{ml}}^2 f^2 / M_f^4 = 1.28$ in the frontal regions. The two fronts are initialized from rest (unbalanced with the horizontal temperature gradients) with small

perturbations on the temperature fields to promote the development of mixed layer instability. The surface is forced by constant wind stress of 0.1 N/m$^2$ in the $x$-direction. Since the MPAS-Ocean simulation domain is doubly periodic in the horizontal, the two fronts move to the right of the wind direction due to Ekman transport. The result is a stable front in which warmer surface water move over on top of the cold water and an unstable front in which both the Ekman-driven convective instability and baroclinic instability occur.

The MPAS-Ocean simulation is first spun up with KPP for 15 days with a time step of 5 min, allowing the mixed layer eddies to develop. Then starting from day 16, three branch runs for 30 hours are conducted, which are the focus of the analysis here. The first run continues to use KPP in all the cells. The second run uses PALM at 8 selected cells and KPP elsewhere, with a PALM domain of 160 m×160 m×120 m and 128×128×120 grid cells, and the same coupling strategy as in the single column test using nudging time scales of $\tau_{\mathrm{LS}}^\theta = 0.5$ h and $\tau_{\mathrm{LS}}^{\boldsymbol{u}} = 5$ h. The third run is the same as the second (i.e., same initial

condition, surface forcing and PALM cells), except there is no coupling between MPAS-Ocean and PALM (i.e., no exchange of large-scale and small-scale forcings). Comparing the former two runs shows the different responses in KPP and PALM to the same surface wind forcing and large-scale forcing due to mixed layer eddies, whereas comparing the latter two shows the impact of the large-scale forcing on the small-scale turbulence in PALM.

Fig. 7 shows a snapshot of the temperature and relative vorticity fields at the beginning of day 16. Note that the unstable

front initially at $y = 46.8$ km has moved to around $y = 25$ km whereas the stable front initially at $y = 15.6$ km has moved to around $y = 45$ km (out and re-entering the domain) due to the Ekman transport towards the negative $y$-direction and the doubly periodic domain. The locations of the eight MPAS-Ocean cells running PALM is marked and labeled for quick reference. For brevity, here we only present the results of four representative cells: cells 1 and 3 locating at both sides of the unstable front, where active mixed layer eddies are developing; cell 7 and 8 locating at both sides of the stable front, with the latter strongly

affected by the Ekman advection of surface warm water.

Fig. 8-10 show the time evolution of the temperature and the zonal and meridional velocities at four locations. It is clearly seen that at cells 1 and 3, both the temperature and momentum are strongly affected by the large-scale forcing due to the mixed layer eddies (panels a, b, d and e in all three figures), where otherwise under constant surface wind forcing and rotation would develop strong inertial oscillation with a period of about 17 hours (panels c and f in Fig. 9 and 10). The similarity between

KPP and PALM at these two locations suggests the dominance of large-scale forcing due to mixed layer eddies in the evolution of temperature and momentum, while small differences are still noticeable suggesting the potential importance of boundary layer turbulence in feeding back to the evolution of mixed layer eddies. At cell 7, the large-scale forcing is relatively small in lack of active mixed layer eddies and Ekman advection of surface warm water. Therefore, the temperature and momentum are similar to the uncoupled PALM simulations (panels g, h and i of all three figures). At cell 8, a second shallower mixed layer

is developed and getting warmer and warmer as a result of the Ekman advection of surface warm water. The temperature is





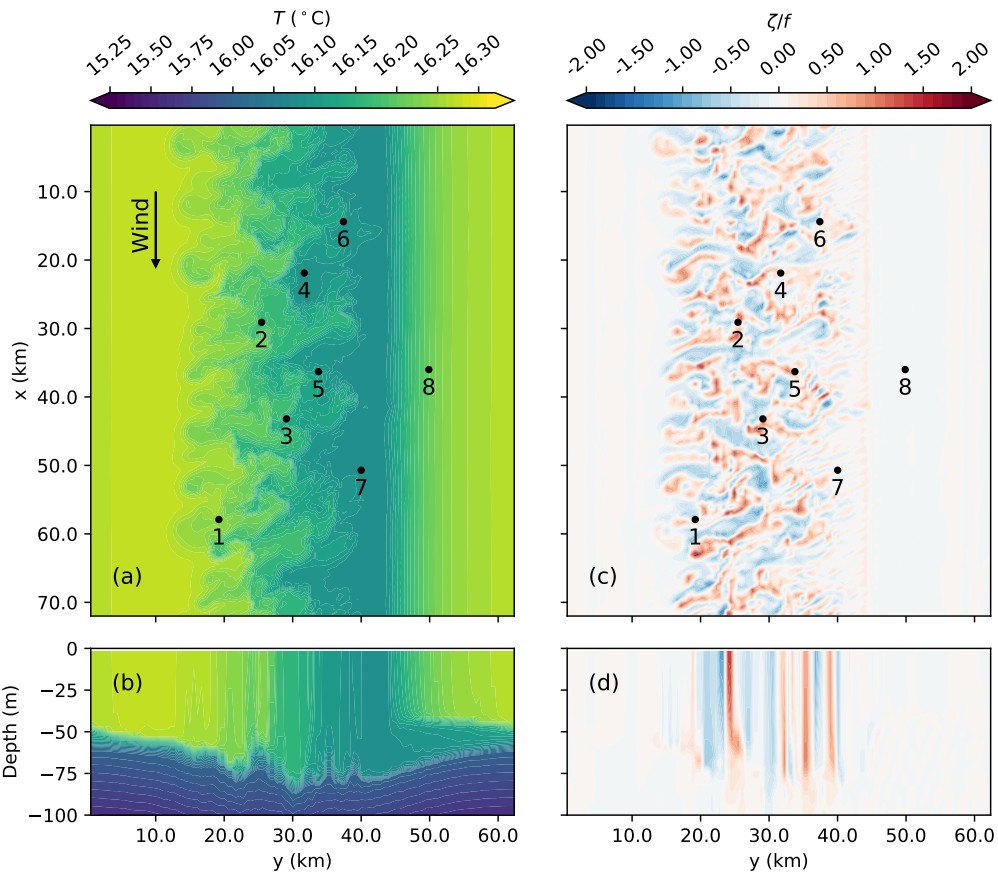

**Figure 7.** Horizontal (a,c) and vertical (b,d) snapshots of the temperature (a,b; $T$ in °C) and relative vorticity normalized by the Coriolis parameter (c,d; $\zeta/f$) at the beginning of day 16 in the mixed layer eddy case. The location of each of the MPAS-Ocean cells running PALM is marked by a black dot and assigned a number for quick reference. The wind is in the $x$-direction as shown by the arrow, with a constant wind stress of 0.1 N/m$^2$. Note that the contour lines for temperature have an interval of 0.05 °C below 16 °C and 0.01 °C above to highlight the eddy structures in the mixed layer.



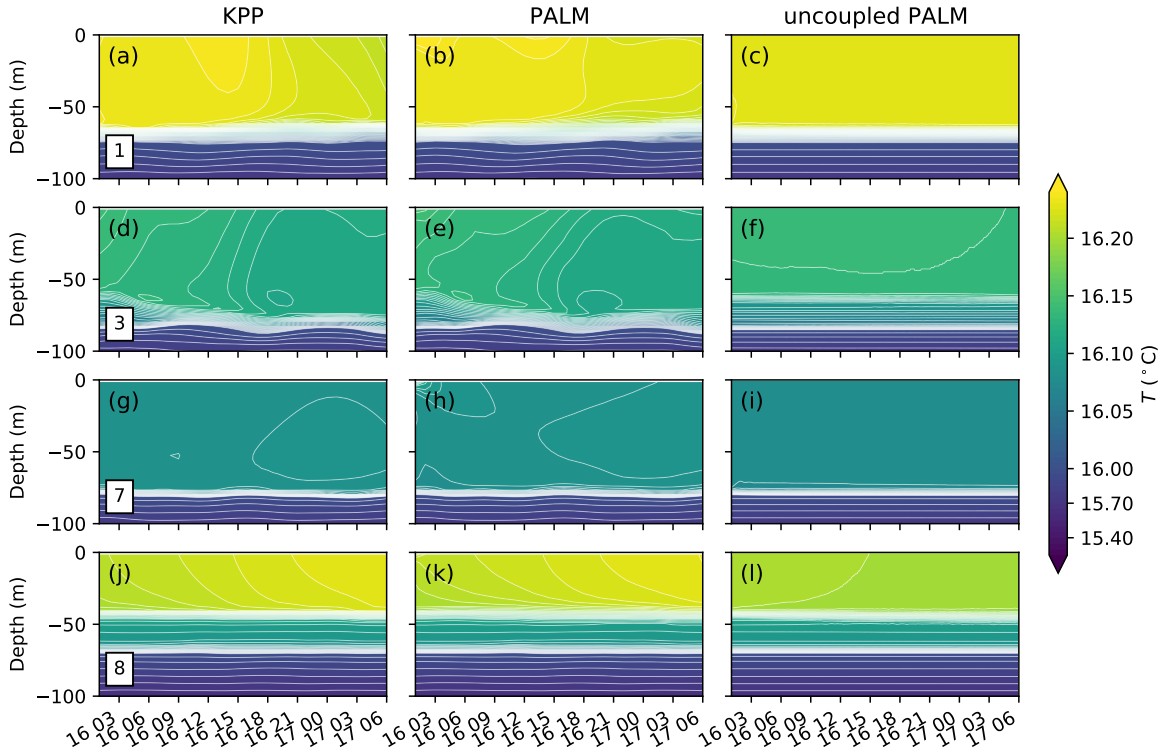

**Figure 8.** Evolution of temperature profiles at the cells 1, 3, 7 and 8 labeled in Fig. 7. Three columns of panels show the profiles in the MPAS-Ocean simulations with KPP (left) and PALM (middle), and the profiles in the uncoupled PALM (right). Note that the contour lines for the temperature have an interval of 0.05 °C below 16 °C and 0.005 °C above to highlight the structures in the mixed layer.

strongly affected by this large-scale forcing (bottom panels of Fig. 8) whereas the momentum is relatively less affected (bottom panels of Fig. 9 and 10).

Finally, we demonstrate the influence of large-scale forcing on the small-scale turbulence statistics by showing the time evolution of the vertical buoyancy flux profile in the three runs in Fig. 11. The buoyancy flux in the KPP run (left panels) is diagnosed from the MPAS-Ocean output of the stratification, the vertical turbulent diffusivity and the non-local fluxes (zero in this case). The results in the coupled and uncoupled PALM runs (middle and right panels) are the sum of the resolved and subgrid-scale buoyancy fluxes in PALM. Consistent with Fig. 8-10, the vertical buoyancy flux is strongly affected by the large-scale forcing due to mixed layer eddies and Ekman-driven convection at cells 1 and 3, but less so at cells 7 and 8. Though qualitatively similar, the vertical buoyancy fluxes in response to the large-scale forcing of mixed layer eddies are quantitatively different between MPAS-Ocean runs with KPP and with PALM, suggesting the potential importance of a better representation of boundary layer turbulence in simulating the mixed layer eddies.

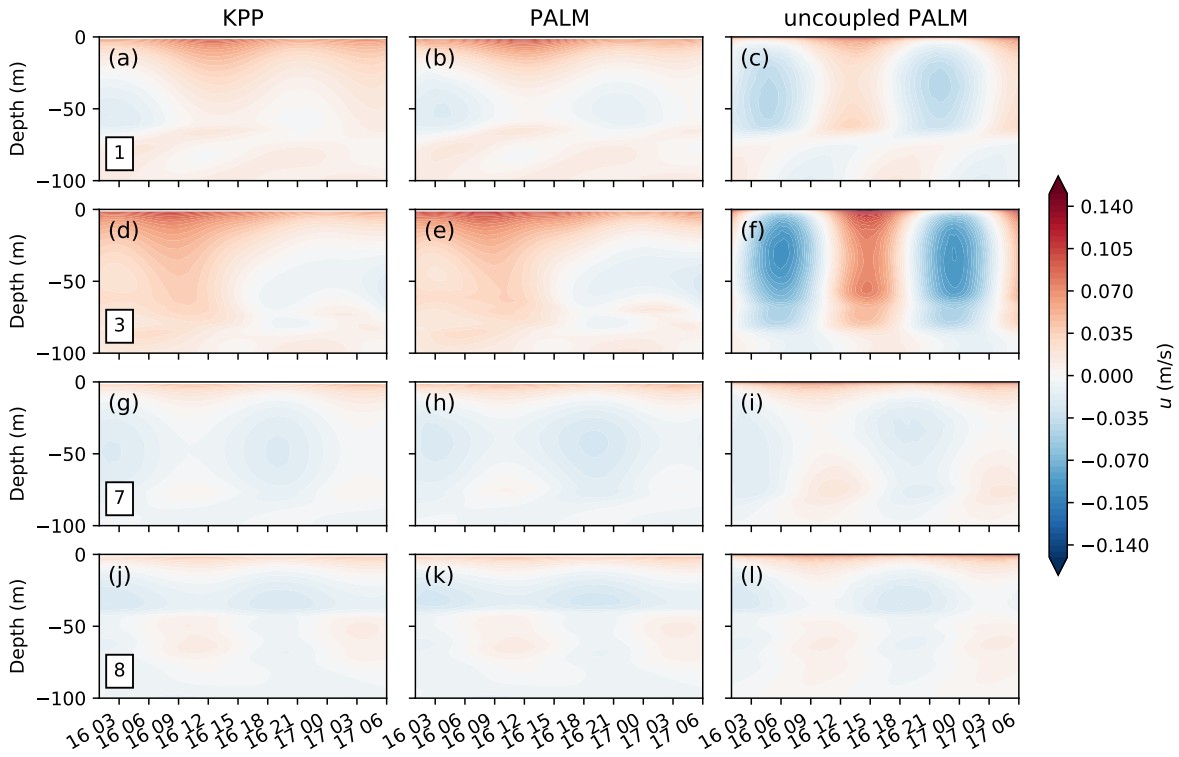

**Figure 9.** Same as Fig. 8, but for the zonal velocity .

Traditionally an OSBL parameterization is often tuned against some forced LES without large-scale forcing, as in the uncoupled PALM case (right panels of Figs. 8-11), whereas a mixed layer eddy parameterization is tuned using GCMs with some OSBL parameterization, as in the KPP case (left panels). The results here suggest that we might need to explicitly

consider the influence of the large-scale processes on the OSBL turbulence and vice versa when tuning the parameters of a parameterization. The coupled MPAS-Ocean and PALM provides a way to do this. It should be noted that here we are showing the differences, rather than quantifying the errors, between the runs with and without the coupling across scales. The latter would require a careful evaluation against an LES simulation of the interactions between mixed layer eddies and the OSBL turbulence (e.g., Hamlington et al., 2014).

**4   Discussion**

Traditionally, the primary goal of embedding a fine-resolution process-resolving model inside a coarse-resolution GCM is to improve the skill of the coarse-resolution GCMs by improving the representation of small-scale processes (thus the term superparameterization). Alternatively, here we show that embedding a high-fidelity, three-dimensional LES in GCMs also

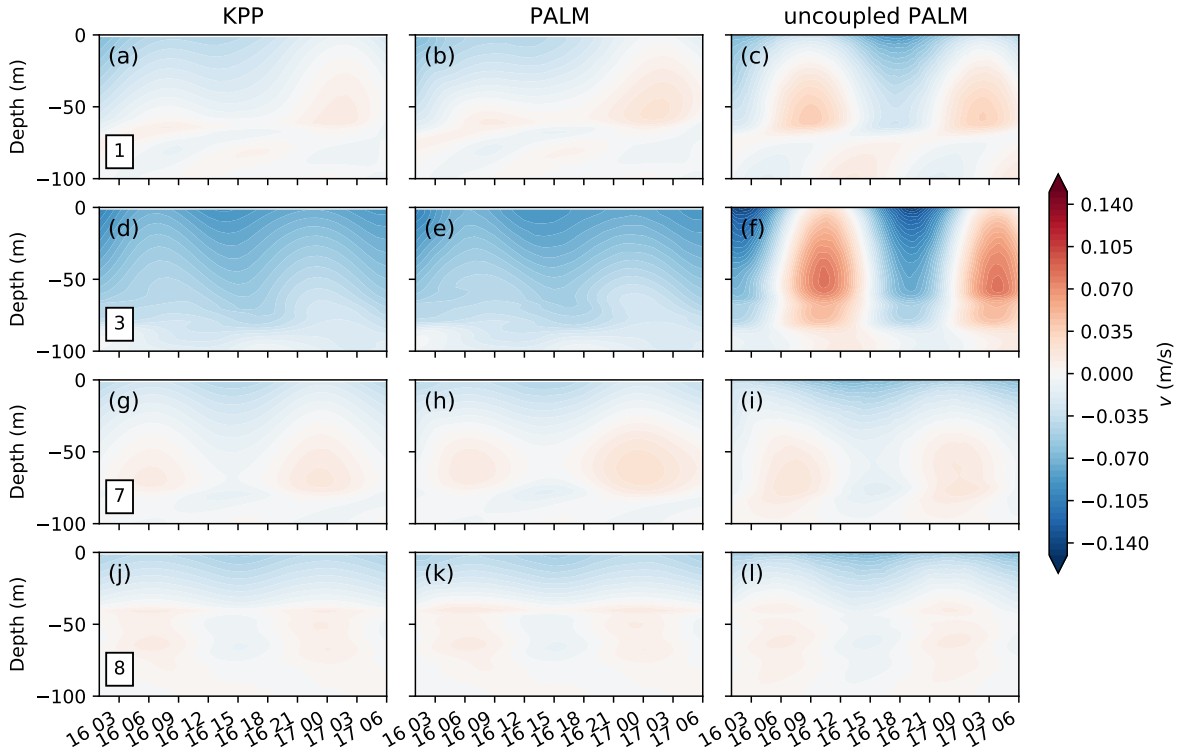

**Figure 10.** Same as Fig. 8, but for the meridional velocity.

provide a useful framework to systematically conduct process studies of turbulent mixing in the OSBL in the context of other

larger-scale processes, such as submesoscale eddies and fronts. In the traditional case, the interests are on the large-scale GCMs

so that the embedded fine-resolution model does not have to be accurate on its own merits, i.e., we only need to represent the

most important effects of the small-scale processes on large-scales. This opens the door for various choices of the embedded

small-scale model with reduced dimension or reduced physics, such as stochastic models (e.g., Grooms and Majda, 2013)

or machine learning models (e.g., Brenowitz and Bretherton, 2018; O'Gorman and Dwyer, 2018). In our case, however, the

interests are also on the small-scale processes. Therefore, a process-resolving model with high-fidelity such as an LES is

required.

One advantage of applying the coupled MPAS-Ocean and PALM to focused process studies, as compared to the traditional

approach of forced LES, is that the small-scale LES is essentially coupled, though rather loosely, with the large-scale GCM.

The effect of large-scale processes, such as advection and lateral mixing, are naturally accounted for at different levels of

completeness depending on the coupling strategy. We can therefore use different coupling strategies to study the interactions

between the large-scale processes and the small-scale turbulent mixing. Additionally, this approach is also much computation-



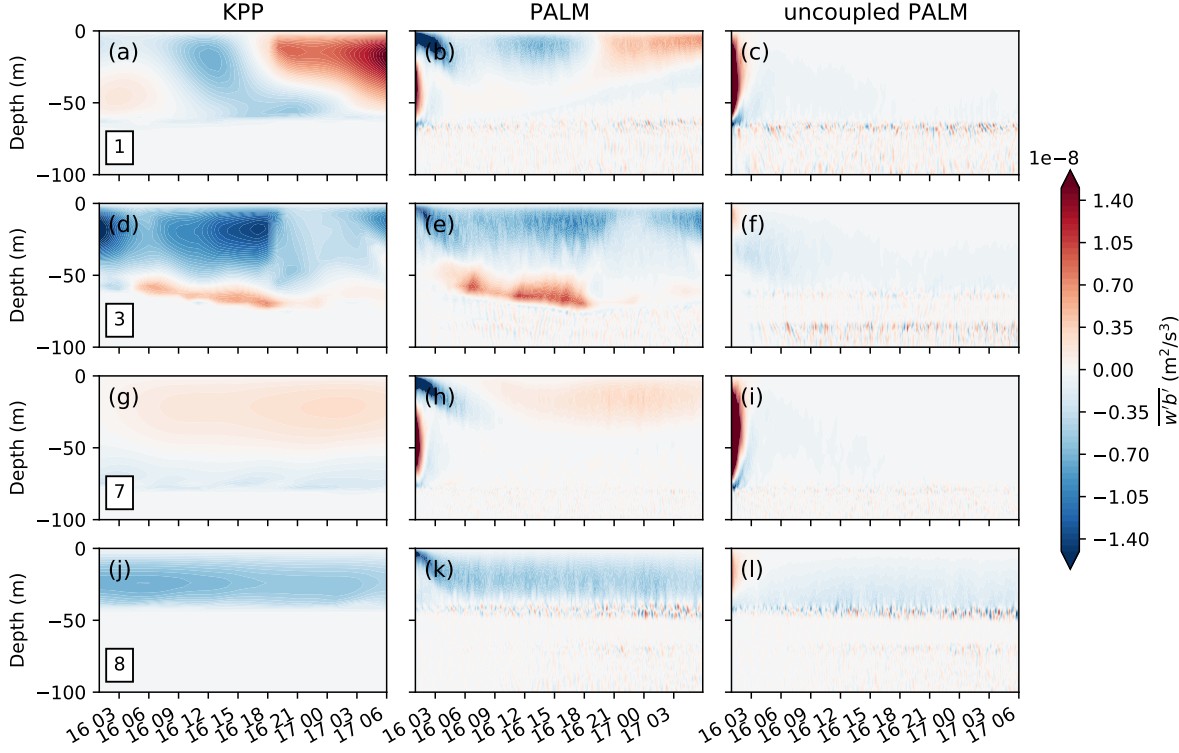

**Figure 11.** Same as Fig. 8, but for the vertical buoyancy flux. Unlike Fig. 8, the middle panels are directly from the PALM output. The large values at the beginning of the simulation for, e.g., (b), (c), (h) and (i) are due to the spin-up of the turbulence in PALM.

ally efficient as compared to an LES on a large domain to resolve all the important processes, which is often required to study the coupling across scales (e.g., Hamlington et al., 2014; Sullivan and McWilliams, 2019).

Given that the direct measurements of turbulent mixing in the ocean is sparse, recent development and tuning of new OSBL
vertical mixing schemes for GCMs rely heavily on forced LES under a variety of different forcing conditions (e.g., Harcourt, 2013, 2015; Li and Fox-Kemper, 2017; Reichl and Li, 2019). Such LES is often idealized without the influences of large-scale processes, assuming that such influences are additive that can be accounted for by other parameterizations in GCMs. As the horizontal resolution of a GCM becomes finer and finer, such assumption may no longer be valid, especially for an OSBL scheme in which the boundary layer depth is derived from a prognostic equation, such as the OSMOSIS schemes (Damerell
et al., 2020). The approach described here will benefit the development of new OSBL schemes by providing an efficient way to generate LES data with the influence of large-scale processes for tuning of the parameters.



## 4.1 Remaining Issues

The coupling between MPAS-Ocean and PALM presented in Section 2.2 currently excludes the effect of horizontal gradient
of large-scale quantities. Horizontal gradient of large-scale quantities is needed in the small-scale dynamics to allow processes
like baroclinic instability (e.g., Bachman and Taylor, 2016) and is essential for the energy transfer from the large-scale to
the small-scale. In the superparameterization literature, as well as the conventional OSBL parameterization literature, it is
often assumed that there is a scale separation between the large-scale processes simulated by the GCMs and the small-scale
turbulent processes simulated by the embedded process-resolving model or represented by the parameterization schemes. As
the horizontal resolution of the GCMs increases, therefore the scales of the resolved process and the subgrid-scale process get
closer, this assumption may start to break. The multiscale modeling approach presented here can by extended to incorporate
such effect by allowing the large-scale forcing terms in Eqs. (19) and (21). Similar approach has been suggested in, e.g.,
Grooms and Julien (2018) for multiscale modeling, as well as applied in LES studies of the frontal regions (e.g., Bachman and
Taylor, 2016; Fan et al., 2018). The tricky part is estimating the horizontal gradient of large-scale quantities, which strongly
depends on the horizontal resolution of the GCMs. Therefore, great caution has to be exercised when interpreting the resolved
horizontal gradients in the coarse-resolution GCMs. In addition, as the size of a grid cell in the GCM becomes similar to the
size of the embedded LES domain, the exchange of lateral fluxes may also need to be explicitly considered. These extensions
to the current approach will be explored in a future study.

Even with the GPU-acceleration technique, running PALM in MPAS-Ocean is still computationally intensive. In the present
setup in the single column test (running PALM on one cell while using KPP in others), the total run time on one CPU and
one GPU is a few hundreds times longer than using KPP in all cells. In the mixed layer eddy test case, running PALM
on 8 cells increases the total run time on 16 CPUs and 8 GPUs by over 40 times. While the computational cost is well
justified for applications on studying the small-scale turbulent mixing, for applications on improving the GCM performance,
the superparameterization approach with high-fidelity and three-dimensional LES embedded in every grid cell of a GCM is
still far from practical. We discuss some possible paths forward to combat the cost of using PALM in MPAS-Ocean in the next
section.

## 4.2 Moving Forward

We have tested the coupled MPAS-Ocean and PALM using some very idealized test cases. The goals of these test cases are to
verify the functionality of the coupled system, which provides the foundation for future extensions and applications to more
realistic and complicated setup. Here we outline a few possible use scenarios that will take advantage of the development in
this study.

### 4.2.1 Effects of Lateral Gradients

As mentioned in the previous section, one obvious step forward is to include the effects of large-scale lateral gradients in
the embedded LES. This allows the development of baroclinic instability in the embedded LES and a direct comparison





with an LES on a larger domain to study the turbulent mixing in the present of large-scale features such as submesoscale
fronts and filaments (e.g., Hamlington et al., 2014; Sullivan and McWilliams, 2019). In addition to significantly reducing the
computational cost as compared to an LES on a large domain, the coupled MPAS-Ocean and PALM also allows one to separate
the contributions of lateral processes and vertical mixing at different scales by combining different domain sizes, resolutions
and coupling strategies for the large-scale and small-scale dynamics. Of course in doing so a horizontal scale separation is
assumed and certain feedback from the small-scales to the large-scales is lost. One step further to address this is to include the
lateral fluxes from the small-scale LES to the large-scale GCM in Eqs. (6) and (8). Another drawback of this setup versus a
large LES is that the maximum large-scale gradients are limited by that can be resolved in the GCM. This can be alleviated by
the flexibility of mesh resolutions in MPAS-Ocean, which will be discussed next.

### 4.2.2  Focused Process Studies

With the capability of MPAS-Ocean to use regionally refined mesh, it is possible to setup simulations for focused process
studies with high-resolution in the study region and low-resolution in the larger background domain. This strategy has already
been adopted to conduct global MPAS-Ocean simulations with refined resolution in the US coast and regional simulations with
a focus on estuaries (personal communication). The coupling with PALM can further extend the refined study region in an
MPAS-Ocean simulation to include non-hydrostatic dynamics. In such applications one can choose to embed PALM inside the
finest grid cells of MPAS-Ocean in the focused regions, where the domain size of PALM can be similar to the grid cell size of
MPAS-Ocean. In this way, it is reasonable to pass the large-scale gradients estimated in the MPAS-Ocean directly to PALM to
allow the development of baroclinic instability and other processes in PALM as discussed above. It would also be interesting
to study the impact of turbulent mixing transitioning from an estuary environment to an open ocean environment using this
coupled system. This is critical for a seamless coastal to global simulation that MPAS-Ocean is targeted at.

Another application is a systematic exploration of the OSBL turbulent mixing using LES under regional and global forcing
conditions with and without the large-scale forcing. Such studies are traditionally conducted by setting up a set of LES using a
range of forcing conditions. One example is LES studies of Langmuir turbulence under hurricane conditions at different phases
and locations (e.g., Reichl et al., 2016; Wang et al., 2018). The coupled MPAS-Ocean and PALM will enable simultaneous
simulations of the hurricane in the GCM and turbulent mixing in the LES, thereby allowing an assessment of the effects of
lateral adjustment versus surface forcing during a hurricane. Such setup also provides useful datasets to assist the development
of ocean turbulent mixing parameterizations in these scenarios.

### 4.2.3  Data Assimilation

Taking advantage of the flexibility of running PALM only on selected locations in MPAS-Ocean, one can also explore the
possibility of improving the simulation results of a GCM by having high-fidelity representations of the turbulent mixing at
only a few locations, borrowing ideas from the data assimilation literature. The high-fidelity representations of the turbulent
mixing can be easily drawn from the coupled LES as shown here. Results of such exploration will also be relevant to incorporate
direct measurements of OSBL turbulent mixing at various platforms such as research vessels and ocean stations.



# 5 Conclusions

In this paper we have outlined the steps and the progress towards a multiscale modeling approach of the ocean surface turbulent mixing by coupling MPAS-Ocean with PALM, following some ideas of the superparameterization approach. However, in
contrast to the traditional superparameterization approach which seeks a way to replace the conventional parameterizations, the goal here is to build towards a flexible framework to better understand the interactions among different processes in the OSBL and thereby providing a pathway to improve the conventional OSBL turbulent mixing parameterizations. For this reason, a high-fidelity and three-dimensional LES is used instead of a simplified model with reduced physics and/or reduced dimension.

To alleviate the computational burden of the embedded LES, we ported a customized version of PALM on GPU using
OpenACC and achieved an overall speedup of 10-16. Although further speedup is possible, the relatively high computational demand of a high-fidelity and three-dimensional LES still prevents a superparameterization-like approach from being practical, in which the LES is running on each grid cell of the GCM. However, it is certainly computationally more favorable than an LES on a large domain.

The flexibility of running LES only on selected grid cells of the GCM, combined with the capability of MPAS-Ocean
using unstructured grid, provides a promising pathway to move forward in simulating the ocean surface turbulent mixing in a multiscale modeling approach. Here we have demonstrated the functionality and potential of this multiscale modeling approach using very simple test cases. As discussed in the previous section, direct process can be made on various applications of this approach, from a regionally focused process study of the OSBL turbulence in the presence of large-scale phenomena, to a systematic exploration of better representations of the small-scale OSBL turbulent mixing in the GCM, perhaps in combination
with the data assimilation and machine learning techniques.

*Code availability.* The source code of the coupled MPAS-Ocean and PALM is available at http://doi.org/10.5281/zenodo.4131134.

*Author contributions.* QL and LVR together designed and implemented the coupling between MPAS-Ocean and PALM, including customizing a version of PALM and porting it on GPU, and prepared the manuscript. QL conducted the simulations and analyzed the results.

*Competing interests.* The authors declare no competing interests.

*Acknowledgements.* Research presented in this article was supported by the Laboratory Directed Research and Development program of Los Alamos National Laboratory under project number 20180549ECR. This research used resources provided by the Los Alamos National Laboratory Institutional Computing Program, which is supported by the U.S. Department of Energy National Nuclear Security Administration



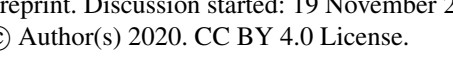 

under Contract No. 89233218CNA000001. This research used resources of the Oak Ridge Leadership Computing Facility, which is a DOE Office of Science User Facility supported under Contract DE-AC05-00OR22725.



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
