# Peer review of "Towards Multiscale Modeling of Ocean Surface Turbulent Mixing Using Coupled MPAS-Ocean v6.3 and PALM v5.0"

_Geoscientific Model Development, 2020_

## Referee Comment (RC1) · Anonymous Referee #1 · 12 Jan 2021

Review of "Towards Multiscale Modeling of Ocean Surface Turbulent Mixing Uxing Coupled MPAS-Ocean v6.3 and PALM v5.0" submitted to Geoscientific Model Development

This paper discusses and presents some preliminary results from efforts to couple a hydrostatic hydrodynamic ocean model (MPAS, the parent model) to a nonhydrostatic horizontally-periodic large eddy simulation (LES) ocean model (PALM, the child model). The PALM model is ported to GPU, which results in a $\sim$10x speedup relative to a single CPU. And, a couple simple test cases are presented.

I find that this paper is a solid and significant contribution to GMD; the approach is novel and has potential value for ocean and climate modeling. However, the test case

results are of limited scientific value; they simply provide preliminary evidence that the implementation is likely correct or close to correct. The methods generally seem reasonable, and the results seem like they flow from the methods. However, the chosen test cases and the methods seem ad hoc and preliminary. In particular, it seems plausible that the approach chosen here is a suboptimal first step, rather than a polished model. I'm particularly concerned about the coupling via nudging and running in coupled mode with different vertical mixing schemes at adjacent grid points; these issues seem scientifically problematic if not technically problematic. The paper is also well written and there are only a few typos, and the software is evidently public, so it is likely to be reproducible, but I did not try to reproduce the results myself. It would be hard to reproduce from the manuscript alone (without the code), but that seems ok to me. Overall, I think this paper is a useful contribution to the literature and should be published. Some specific comments are below.

Specific comments: It is hard for me to interpret the practical implications of porting LES to GPU in the context of existing system architectures. The small test cases presented here don't really highlight the advantages of the approach at scale, i.e. the examples are all cheap/small calculations. The implementation is also not exactly clear. Does each LES grid point use 1 CPU and 1 GPU? Is the application limited by the number of GPUs on a typical node?

There seems to be too much scientific background material and the motivation is confusing. For example, there is much ado about "superparameterization," but the suggestion is that using this coupled model with LES in each grid cell as a superparameterization is likely to be infeasible (e.g. L70). I don't think that this general suggestion can be accepted or rejected based on the benchmarking and small number of examples and results in the paper. However, I agree that the practical value of such a model is its ability to provide LES dynamics in the context of larger-scale ocean dynamics, as the authors suggest. I think the paper should focus on this latter motivation, and on one-way MPAS->LES coupling. Reduce some of the discussion of superparameterization

and two-way coupling to make the manuscript more focused.

The coupling between parent and child is ad hoc. For example, MPAS -> LES mean profile is a nudging with a timescale of 30 min for tracers and 5 hours for momentum. There is some discussion of sensitivity to the nudging timescale, but there are alternative approaches that are not explored. For example, it may better to apply the dynamical tendencies from MPAS to the LES mean profiles (excluding vertical mixing terms), rather than or in addition to nudging the mean profiles to the parent model. In addition, the omission of mean lateral gradients in the LES significantly modifies the turbulence energetics where gradients are strong (e.g. at fronts) in the second example (e.g. Bachman et al. 2017). These explicit lateral gradient effects are missed with the nudging (and the forcing by large scale tendencies).

Bachman, S. D., Fox-Kemper, B., Taylor, J. R., & Thomas, L. N. (2017). Parameterization of frontal symmetric instabilities. I: Theory for resolved fronts. Ocean Modelling, 109, 72-95

The inconsistency between adjacent grid points (some points with KPP and some points with LES) makes the two-way coupling too problematic to use without further exploration. I think it would be better to start by exploring science questions with only one-way parent-to-child coupling to avoid this problem as a starting point. That is, use KPP at all points to advance the parent model. See my first specific point above about the background on superparameterization.

Please clarify how surface fluxes calculated in the two models. Are there feedbacks between ocean state and atmosphere, as in bulk flux algorithms, that lead to inconsistencies in the fluxes between grid points with KPP and grid points with LES?
* * *

---

## Referee Comment (RC2) · Walter Hannah (Referee) · 25 Jan 2021

**Towards Multiscale Modeling of Ocean Surface Turbulent Mixing**
**Using Coupled MPAS-Ocean v6.3 and PALM v5.0**

Qing Li and Luke Van Roekel

This paper discusses an implementation of super-parameterization (SP) in an ocean model. The embedded small scale model is ported to run on GPUs with openACC and a notable ~10x speed up is reported. A series of simulations is presented to explore the impact of using SP in limited regions.

I am not an ocean modeller, so I don't have much to say about the experiments in the latter part of the paper, but they seem reasonably insightful. However, I have worked a lot with super-parameterization in the atmosphere and I take issue with the discussion of the SP implementation presented in this paper. I feel that the authors have erroneously characterized the paradigm. I have outlined my concerns about this below, along with a specific comment about the estimation of GPU speedup. Other than these concerns the paper is very well written.

**Mischaracterization of super-parameterization**

First off, on a semantic note, I prefer the term "multi-scale modelling framework" (MMF) over "super-parameterization" (SP). I used to consider these as interchangeable terms, but over time I have found that the notion of "replacing parameterizations" with a small-scale model often leads to misunderstandings. I feel "MMF" is the more accurate term since two distinct models that cover different scale ranges are being "coupled" together, which is different than using one model to simply replace the low-order parameterization tendencies of the other.

A key aspect of the MMF/SP idea, going back to the original work by Wojciech Grabowski, is that the large-scale and small-scale models are "tightly" coupled, *which is not the same as nudging*, even though the forcing and feedback terms may resemble nudging tendencies. On line 217 of the current manuscript the authors state that the requirement of mean state equality between the two models is enforced by either nudging the small-scale model or simply replacing the small-scale mean state with the large-scale mean state. I went back to the Khairdinov et al. (2005) paper cited in that sentence, and it makes a single mention of "nudging" when talking about the uncoupled 2D momentum field, but that paper never describes the coupling method as "nudging". The use of "nudging" implies that the small-scale model has no ability to impact the GCM state variables, but this is an erroneous characterization of traditional super-parameterization. As far as I know, nudging has *never* been used for SP in an atmosphere model outside a few special cases like the Q3D model of Jung and Arakawa (2014) and maybe the earliest implementation by Grabowski.

The notion of tight coupling means that the smallest scale of the large-scale model is enforced to be equal to the largest scale of the small-scale model (i.e. domain mean) through the formulation of the forcing tendencies in both directions. This keeps the models synchronized across the scale gap (see diagram).

[Figure]

Another way to think about the coupling strategy is that it is very scale selective. This is why Grabowski (2004) formulates the forcing/feedback tendencies to occur specifically on these scales. Having this mindset, I was very confused when reading section 2.1 of the current manuscript, which seems to formulate the coupling *at the smallest scale of the small-scale model*. The authors ultimately use a nudged version of the classic SP formulation, so I fail to see the relevance of section 2.1 to the manuscript, outside of the discussion of the impact of gradients on the small-scale processes.

On that note, the idea of explicitly including the effect of large-scale gradients in the small-scale model is very interesting. The authors' discussion of this seems to be in the context of equations (19) and (21), but following from my comments above, these equations are not consistent with the traditional SP formulation in which the coupling occurs at the largest scale of the small-scale model. Equations (19) and (20) imply that the coupling is valid on the smallest scale of the small-scale model, which is a very intriguing concept. However, It is difficult to imagine how the concept of "tight coupling" could be applied with this approach, so a nudging framework would probably be needed.

The idea of representing gradients in the small scale model would also require overcoming the periodic boundary conditions. The authors don't really address how this would be possible, except for a mention of perhaps needing to exchange lateral boundary fluxes between the different instances of the small-scale model. It's worth noting that this idea is problematic for performance reasons. It seems that this would require a huge increase in inter-process communication, and would certainly ruin any potential GPU speedup due to the extra GPU/CPU data exchange. I don't think the authors have really thought through these issues based on the discussion in the manuscript.

In summary, I think the authors need to revisit their description of the method used to couple the two models with special attention paid to the scale at which the coupling occurs, as well as a more accurate characterization of previous work.

**GPU Speedup**

When estimating the GPU speedup for E3SM-MMF we often use an entire Summit node (2 CPUs vs 6 GPUs), but we still have ongoing discussions about how to make the CPU vs GPU comparison "fair". I believe our argument for using (2 CPUS with 42 MPI tasks) vs (2 CPUS + 6 GPUs with 12 MPI tasks) is based on power consumption, along with some subtle aspects of our specific configuration. We also often estimate GPU speedup with standalone versions of the small-scale model to isolate its performance from the large-scale model. Obviously, estimating the model throughput "per watt" would be a much more ideal way to measure speed-up for these different configurations, but that is difficult to obtain.

For the estimate of GPU speedup for PALM, I think mentioning these concerns would be a nice addition to the discussion. Also the number of MPI tasks is important to mention.

- Walter Hannah

**References**

Grabowski, W. W., 2004: An improved framework for superparameterization. J. Atmos. Sci., 61 , 1940–1952.

Khairoutdinov, M. F., D. A. Randall, and C. DeMott (2005), Simulations of the atmospheric general circulation using a cloud-resolving model as a superparameterization of physical processes, J. Atmos. Sci., 62, 2136–2154.

Jung, J.-H., and Arakawa, A. (2014), Modeling the moist-convective atmosphere with a Quasi-3-D Multiscale Modeling Framework (Q3D MMF), J. Adv. Model. Earth Syst., 6, 185– 205, doi:10.1002/2013MS000295.

---

## Author Comment (AC1) · 19 Feb 2021

**Response to Reviewers**

We thank all the reviewers for their very helpful comments and suggestions! All the comments and suggestions were seriously considered and the manuscript has been revised accordingly.

We have rewritten the introduction section to be more focused on the background and the motivations of exploring multiscale modeling framework for the ocean surface turbulent mixing. We have removed a big part of the review and discussions of the superparameterization literature that are not directly relevant to the present study. The differences between our approach and the traditional superparameterization approach are also clarified where appropriate.

We have also revised a big part of the methods section. In particular, the original Section 2.1 is now merged with Section 2.2 and streamlined to focus more on the coupling approach used in this study. Hopefully this will reduce the confusions that may have been caused by the unnecessary equations and discussions.

A point-by-point response to each reviewer is detailed in the following. All line numbers mentioned in this response correspond to the revised manuscript.

**1. Response to reviewer 1**

Review of "Towards Multiscale Modeling of Ocean Surface Turbulent Mixing Uxing Coupled MPAS-Ocean v6.3 and PALM v5.0" submitted to Geoscientific Model Development

This paper discusses and presents some preliminary results from efforts to couple a hydrostatic hydrodynamic ocean model (MPAS, the parent model) to a nonhydrostatic horizontally-periodic large eddy simulation (LES) ocean model (PALM, the child model). The PALM model is ported to GPU, which results in a ~10x speedup relative to a single CPU. And, a couple simple test cases are presented.

I find that this paper is a solid and significant contribution to GMD; the approach is novel and has potential value for ocean and climate modeling. However, the test case results are of limited scientific value; they simply provide preliminary evidence that the implementation is likely correct or close to correct. The methods generally seem reasonable, and the results seem like they flow from the methods. However, the chosen test cases and the methods seem ad hoc and preliminary. In particular, it seems plausible that the approach chosen here is a suboptimal first step, rather than a polished model. I'm particularly concerned about the coupling via nudging and running in coupled mode with different vertical mixing schemes at adjacent grid points; these issues seem scientifically problematic if not technically problematic. The paper is also well written and there are only a few typos, and the software is evidently public, so it is likely to be reproducible, but I did not try to reproduce the results myself. It would be hard to reproduce from

the manuscript alone (without the code), but that seems ok to me. Overall, I think this paper is a useful contribution to the literature and should be published. Some specific comments are below.

Thanks for these comments! These are really good points.

The main purpose of this paper is to explore new approaches applying the multiscale modeling framework to study the ocean surface turbulent mixing. Therefore, the simple test cases are chosen to evaluate the coupling strategies, and to expose potential problems for future improvement, rather than to address a certain scientific question using this coupled model yet, which will be the natural next steps as discussed in the discussion section.

The nudging part of the coupling is a simple choice to start, representing an initial step building towards a more complete multiscale modeling approach. The flexibility of using different relaxation time scales in the relaxing terms allows us to explore different levels of tightness in the coupling (e.g., the coupling time scale, coupling only between tracers vs both tracers and momentum, etc.). The choice of running LES only on selected GCM grid points allows focused process studies at particular regions of interest without the computational burden of running LES on all GCM grid points. We agree that running LES on certain GCM grid points coupled with different vertical mixing schemes at adjacent grid points is somewhat problematic, as illustrated and discussed using the single column test case in Section 3.1. Issues like this are exactly the things we want to explore and discuss in this technical paper.

Specific comments: It is hard for me to interpret the practical implications of porting LES to GPU in the context of existing system architectures. The small test cases presented here don't really highlight the advantages of the approach at scale, i.e. the examples are all cheap/small calculations. The implementation is also not exactly clear. Does each LES grid point use 1 CPU and 1 GPU? Is the application limited by the number of GPUs on a typical node?

The size of the problem that LES solves in these test cases is indeed relatively small. But still, they are much more computationally expensive than using KPP. To make it practical to advance the parent model MPAS-Ocean, the LES has to be as fast as possible. In this sense, a speed-up of over 10x is significant and certainly helpful. A speedup of 10x is also the somewhat accepted minimum requirement to be effective using the GPU, given the increased power consumption. As pointed out by the other reviewer, a measure of model throughput per watt of energy consumption would be the ideal way to show the benefit of porting LES to GPU. However, that measure is hard to obtain.

Yes, each LES grid point uses 1 CPU and 1 GPU. Since there is no coupling between different LES instances, these LES instances on different grid points can run in parallel using multiple GPUs. But we can also run multiple LES instances on a single GPU. The number of LES instances we can run on a single node is limited by the memory available to GPUs, not the number of GPUs. The setup is flexible in the sense that we can allocate the available GPU

resources between the size of the problem for each LES grid point and the number of LES grid points.

We have revised Section 2.3 (now Section 2.2) to clarify on the implementation porting PALM on GPU and the interpretation of the GPU speedup measured by the runtime. We also added some discussions on alternative measures of the GPU speedup following the suggestion of the other reviewer.

There seems to be too much scientific background material and the motivation is confusing. For example, there is much ado about "superparameterization," but the suggestion is that using this coupled model with LES in each grid cell as a superparameterization is likely to be infeasible (e.g. L70). I don't think that this general suggestion can be accepted or rejected based on the benchmarking and small number of examples and results in the paper. However, I agree that the practical value of such a model is its ability to provide LES dynamics in the context of larger-scale ocean dynamics, as the authors suggest. I think the paper should focus on this latter motivation, and on one-way MPAS->LES coupling. Reduce some of the discussion of superparameterization and two-way coupling to make the manuscript more focused.

The purpose of introducing superparameterization is to highlight the relation and in particular the differences between our approach and the superparameterization approach. At the beginning of this project there was an attempt to explore a superparameterization-like approach, hoping the GPU acceleration would make it computationally feasible to embed an LES in every MPAS-Ocean grid point. But we quickly realized that even with the GPU acceleration, this is far from practical. We discussed this issue in Section 4.1.

To make the motivations and the background information of this manuscript more focused, the introduction section has been rewritten. The background information on superparameterization is now reduced to avoid possible confusions.

The coupling between parent and child is ad hoc. For example, MPAS -> LES mean profile is a nudging with a timescale of 30 min for tracers and 5 hours for momentum. There is some discussion of sensitivity to the nudging timescale, but there are alternative approaches that are not explored. For example, it may better to apply the dynamical tendencies from MPAS to the LES mean profiles (excluding vertical mixing terms), rather than or in addition to nudging the mean profiles to the parent model. In addition, the omission of mean lateral gradients in the LES significantly modifies the turbulence energetics where gradients are strong (e.g. at fronts) in the second example (e.g. Bachman et al. 2017). These explicit lateral gradient effects are missed with the nudging (and the forcing by large scale tendencies).

Bachman, S. D., Fox-Kemper, B., Taylor, J. R., & Thomas, L. N. (2017). Parameterization of frontal symmetric instabilities. I: Theory for resolved fronts. Ocean Modelling, 109, 72-95

The inconsistency between adjacent grid points (some points with KPP and some points with LES) makes the two-way coupling too problematic to use without further exploration. I think it would be better to start by exploring science questions with only one-way parent-to-child coupling to avoid this problem as a starting point. That is, use KPP at all points to advance the parent model. See my first specific point above about the background on superparameterization.

Exploring different coupling strategies is one of the main purposes of this manuscript and we did try a few different approaches. Applying the dynamical tendencies from MPAS-Ocean to LES is equivalent to relaxing the LES mean fields to the MPAS-Ocean fields with a relaxation time scale of the MPAS-Ocean time step (30 min in this case).

One way coupling is certainly one useful option. But we also want to explore the feasibility of two-way coupling with different representations of the ocean vertical mixing at different grid points. The issue of two-way coupling with inconsistent representations of the vertical mixing between adjacent grid points is actually a result of this practice we would like to report, as discussed in the context of the single column test in Section 3.1. It is interesting to see whether we can incorporate useful information from LES at only selected locations. Again, the main purpose of this technical paper is not to address some specific scientific questions but rather to develop and validate tools that may be useful in future studies. One way coupling with lateral gradient effects is certainly a natural step forward as discussed in Section 4.2.

Please clarify how surface fluxes calculated in the two models. Are there feedbacks between ocean state and atmosphere, as in bulk flux algorithms, that lead to inconsistencies in the fluxes between grid points with KPP and grid points with LES?

Unlike the ocean simulations forced by an atmospheric dataset (e.g., CORE-II or JRA55-do), the surface fluxes are directly applied as boundary conditions. There is no feedback between the ocean state and atmosphere as in the bulk flux algorithms. So the fluxes are consistent between grid points with KPP and with LES. This point is now clarified in the discussion of Fig. 6 (Lines 350-352).

**2. Response to reviewer 2**

This paper discusses an implementation of super-parameterization (SP) in an ocean model. The embedded small scale model is ported to run on GPUs with openACC and a notable ~10x speed up is reported. A series of simulations is presented to explore the impact of using SP in limited regions.

I am not an ocean modeller, so I don't have much to say about the experiments in the latter part of the paper, but they seem reasonably insightful. However, I have worked a lot with super-parameterization in the atmosphere and I take issue with the discussion of the SP

implementation presented in this paper. I feel that the authors have erroneously characterized the paradigm. I have outlined my concerns about this below, along with a specific comment about the estimation of GPU speedup. Other than these concerns the paper is very well written.

Thanks a lot for pointing these out! We have revised the methods section that hopefully has clarified these points.

**Mischaracterization of super-parameterization**

First off, on a semantic note, I prefer the term "multi-scale modelling framework" (MMF) over "super-parameterization" (SP). I used to consider these as interchangeable terms, but over time I have found that the notion of "replacing parameterizations" with a small-scale model often leads to misunderstandings. I feel "MMF" is the more accurate term since two distinct models that cover different scale ranges are being "coupled" together, which is different than using one model to simply replace the low-order parameterization tendencies of the other.

Agreed. We have revised the manuscript, in particular the introduction and methods sections to clarify the differences between our multi-scale modeling approach and the superparameterization approach.

A key aspect of the MMF/SP idea, going back to the original work by Wojciech Grabowski, is that the large-scale and small-scale models are "tightly" coupled, which is not the same as nudging, even though the forcing and feedback terms may resemble nudging tendencies. On line 217 of the current manuscript the authors state that the requirement of mean state equality between the two models is enforced by either nudging the small-scale model or simply replacing the small-scale mean state with the large-scale mean state. I went back to the Khairdinov et al. (2005) paper cited in that sentence, and it makes a single mention of "nudging" when talking about the uncoupled 2D momentum field, but that paper never describes the coupling method as "nudging". The use of "nudging" implies that the small-scale model has no ability to impact the GCM state variables, but this is an erroneous characterization of traditional super-parameterization. As far as I know, nudging has never been used for SP in an atmosphere model outside a few special cases like the Q3D model of Jung and Arakawa (2014) and maybe the earliest implementation by Grabowski.

The notion of tight coupling means that the smallest scale of the large-scale model is enforced to be equal to the largest scale of the small-scale model (i.e. domain mean) through the formulation of the forcing tendencies in both directions. This keeps the models synchronized across the scale gap (see diagram).

Indeed, the flexible coupling strategy described in this manuscript is one of the main differences between our multiscale modeling approach and the traditional superparameterization approach. We are using a nudging term here, in particular allowing different relaxation time scales than the

MPAS-Ocean time step, because it gives us more flexibility of choosing how tight the small scale dynamics and large scale dynamics are coupled. The option to use different relaxation time for tracers and momentum allows more flexibility due to the staggered MPAS-Ocean grid. This point is now clarified in the discussion following the relaxing terms (Lines 146-151).

Another way to think about the coupling strategy is that it is very scale selective. This is why Grabowski (2004) formulates the forcing/feedback tendencies to occur specifically on these scales. Having this mindset, I was very confused when reading section 2.1 of the current manuscript, which seems to formulate the coupling at the smallest scale of the small-scale model. The authors ultimately use a nudged version of the classic SP formulation, so I fail to see the relevance of section 2.1 to the manuscript, outside of the discussion of the impact of gradients on the small-scale processes.

On that note, the idea of explicitly including the effect of large-scale gradients in the small-scale model is very interesting. The authors' discussion of this seems to be in the context of equations (19) and (21), but following from my comments above, these equations are not consistent with the traditional SP formulation in which the coupling occurs at the largest scale of the small-scale model. Equations (19) and (20) imply that the coupling is valid on the smallest scale of the small-scale model, which is a very intriguing concept. However, It is difficult to imagine how the concept of "tight coupling" could be applied with this approach, so a nudging framework would probably be needed.

The scale of the coupling indeed occurs at the largest scale of the embedded LES. For the effects of small-scales on large-scales, we are computing the domain averaged statistics (specifically the convergence of the vertical fluxes) from the embedded LES and assuming these statistics represent the statistics of the missing subgrid-scale physics in the parent model MPAS-Ocean. For the effects of large-scale on small-scale, we are enforcing the horizontally domain averaged LES fields to match the large-scale MPAS-Ocean fields (originally Eqs. (24) and (25), now Eqs. (7) and (8)), which is similar to the superparameterization approach although allowing different relaxation time scales than the MPAS-Ocean time step so that the coupling is not tight.

We realize that the equations and discussions in Section 2.1 were quite confusing. Therefore, we have removed a big part of the equations and discussions in Section 2.1 and merged it with Section 2.2, keeping only the equations that are necessary to illustrate the coupling between MPAS-Ocean and PALM in this study. Hopefully this has clarified our coupling approach.

The idea of representing gradients in the small scale model would also require overcoming the periodic boundary conditions. The authors don't really address how this would be possible, except for a mention of perhaps needing to exchange lateral boundary fluxes between the different instances of the small-scale model. It's worth noting that this idea is problematic for performance reasons. It seems that this would require a huge increase in inter-process

communication, and would certainly ruin any potential GPU speedup due to the extra GPU/CPU data exchange. I don't think the authors have really thought through these issues based on the discussion in the manuscript.

In the embedded domain for the LES, the large-scale lateral gradients can be assumed to be constant, both in time during a coupling period (while time stepping the LES) and in space in the horizontal directions across the domain. The constant lateral gradients can be treated as the background fields. Then we can solve the LES for the perturbations around the background fields, and the perturbations can be assumed to be periodic. Similar technique has already been used in LES studies of turbulent mixing in the context of background buoyancy gradient (in thermal wind balance with the background velocity shear) in ocean frontal regions (e.g., Backman and Taylor, 2016). This is now clarified in the discussion of future work to account for the effects of large-scale lateral gradients in Section 4.2 (Lines 480-484).

Therefore, passing the information of lateral gradients to the embedded domain is similar to the present setup of passing large-scale forcing terms and wouldn't cause too much additional data exchange between the CPU and GPU. The data exchange only occurs at the beginning of each MPAS-Ocean time step. Comparatively, time stepping the LES within each MPAS-Ocean time step is much more computationally expensive. So the GPU speedup is also relevant.

As discussed in Section 4.1, one issue of estimating the large-scale lateral gradients from the MPAS-Ocean fields is the coarse resolution. The best we can get is the grid scale gradients in MPAS-Ocean. However, the flexible configuration described in this manuscript allows us to choose the setup that minimizes the effect of this issue. For example, a sensible use case that accounts for the effect of large-scale lateral gradients on the small scale turbulent mixing is to run MPAS-Ocean with regionally refined mesh and embed LES in the finest grid cells for a focused process study (Section 4.2 in Lines 493-502).

Bachman, S. D., & Taylor, J. R. (2016). Numerical simulations of the equilibrium between eddy-induced restratification and vertical mixing. Journal of Physical Oceanography, 46(3), 919–935. https://doi.org/10.1175/JPO-D-15-0110.1

In summary, I think the authors need to revisit their description of the method used to couple the two models with special attention paid to the scale at which the coupling occurs, as well as a more accurate characterization of previous work.

Thanks for these very helpful comments! A large part of the introduction and the methods sections have now been rewritten to clarify these issues.

**GPU Speedup**

When estimating the GPU speedup for E3SM-MMF we often use an entire Summit node (2 CPUs vs 6 GPUs), but we still have ongoing discussions about how to make the CPU vs GPU comparison "fair". I believe our argument for using (2 CPUS with 42 MPI tasks) vs (2 CPUS + 6 GPUs with 12 MPI tasks) is based on power consumption, along with some subtle aspects of our specific configuration. We also often estimate GPU speedup with standalone versions of the small-scale model to isolate its performance from the large-scale model. Obviously, estimating the model throughput "per watt" would be a much more ideal way to measure speed-up for these different configurations, but that is difficult to obtain.

For the estimate of GPU speedup for PALM, I think mentioning these concerns would be a nice addition to the discussion. Also the number of MPI tasks is important to mention.

We totally agree that a measure of the model throughput per watt of energy consumption would be a much better way to show the benefit of porting PALM on GPU. For our purpose here we are using 1 MPI task on both CPU and GPU. Basically the reason to measure the speedup in the run time with 1 MPI is the following. Since each PALM instance is associated with one MPAS-Ocean grid cell, if we were running the coupled system all on CPU, the PALM instance would be run in serial with only 1 MPI task. So this speedup in run time reflects what we can gain from offloading the PALM instance on GPU in the coupled MPAS-Ocean and PALM framework, though it may not be fair in the sense of energy consumption. We have added some discussion on the measure of the speedup and benefits of porting code on GPU following the reviewer's suggestion, as well as the reasoning of why measuring the speedup in model run time is helpful for our purposes (Lines 251-256).